# CD8+ T cell priming that is required for curative intratumorally anchored anti-4-1BB immunotherapy is constrained by Tregs

Joseph R. Palmeri [1,2], Brianna M. Lax[1,2], Joshua M. Peters [3,4],
Lauren Duhamel [1,3], Jordan A. Stinson[1,3], Luciano Santollani[1,2], Emi A. Lutz [1,3],
William Pinney III[1,3], Bryan D. Bryson [3,4] & K. Dane Wittrup [1,2,3] ✉

Although co-stimulation of T cells with agonist antibodies targeting 4-1BB (CD137) improves antitumor immune responses in preclinical studies, clinical success has been limited by on-target, off-tumor activity. Here, we report the development of a tumor-anchored α4-1BB agonist (α4-1BB-LAIR), which consists of a α4-1BB antibody fused to the collagen-binding protein LAIR. While combination treatment with an antitumor antibody (TA99) shows only modest efficacy, simultaneous depletion of CD4+ T cells boosts cure rates to over 90% of mice. Mechanistically, this synergy depends on αCD4 eliminating tumor draining lymph node regulatory T cells, resulting in priming and activation of CD8+ T cells which then infiltrate the tumor microenvironment. The cytotoxic program of these newly primed CD8+ T cells is then supported by the combined effect of TA99 and α4-1BB-LAIR. The combination of TA99 and α4-1BB-LAIR with a clinically approved αCTLA-4 antibody known for enhancing T cell priming results in equivalent cure rates, which validates the mechanistic principle, while the addition of αCTLA-4 also generates robust immunological memory against secondary tumor rechallenge. Thus, our study establishes the proof of principle for a clinically translatable cancer immunotherapy.

The use of monoclonal antibodies to perturb immune cell signaling networks and improve anti-cancer immune responses has gained increased attention in recent years[1]. Checkpoint blockade therapy with antagonistic antibodies is safe and efficacious, but agonistic antibodies against targets such as 4-1BB, OX40, GITR, and ICOS have proven to exhibit impractically narrow therapeutic windows due to on-target, off-tumor toxicity[2–5].

4-1BB (also known as CD137 or TNFRSF9) is expressed primarily on activated CD8+ and CD4+ T cells, including CD4+ regulatory T cells (Treg), and natural killer (NK) cells, and is a promising target for agonist antibodies[6–10]. Signaling through 4-1BB in CD8+ T cells leads to proliferation, enhanced survival, cytokine production, improved memory formation, and altered metabolism[11–15]. Treating mice with

agonist α4-1BB antibodies as a monotherapy or in combination therapies is highly efficacious in several preclinical mouse cancer models[16,17]. However, toxicity has hampered the clinical translation of such antibodies, with lethal liver toxicities reported in early phase 2 trials of Urelumab, the first α4-1BB agonistic antibody to enter the clinic[18]. At reduced doses which do not elicit dose-limiting toxicities (DLT), little to no clinical efficacy has been reported[18]. Utomilumab, the second α4-1BB agonist to enter the clinic, is well tolerated but is a much weaker agonist and has little clinical activity[19,20]. Given the difficulty of uncoupling toxicity from clinical activity with systemically administered agonists, recent development around this target has focused on engineering antibodies with tumor-specific activity[21]. This includes several bispecific antibodies, with one arm targeting 4-1BB and the

[1]Koch Institute for Integrative Cancer Research, Massachusetts Institute of Technology (MIT), Cambridge, MA, USA. [2]Department of Chemical Engineering, Massachusetts Institute of Technology (MIT), Cambridge, MA, USA. [3]Department of Biological Engineering, Massachusetts Institute of Technology (MIT), Cambridge, MA, USA. [4]Ragon Institute of MGH, MIT, and Harvard, Cambridge, MA, USA. ✉e-mail: wittrup@mit.edu

other targeting either tumor-specific antigens or PD-L1, α4-1BB antibodies that bind only in tumor-specific niches, such as high ATP concentrations, or pro-drug α4-1BB antibodies where the binding domain of the antibody is shielded by a peptide "mask" that is cleaved by tumor specific proteases[22–26].

Alternatively, our group and others have demonstrated the utility of using collagen-binding strategies to anchor immunotherapy payloads to the tumor microenvironment[27–35]. Collagen is a desirable target for localization due to its abundance in the tumor microenvironment (TME)[36]. By directly fusing collagen binding domains to cytokines and chemokines or chemical conjugation of collagen binding peptides to αCTLA-4 and αCD40 antibodies, intratumoral administration of these therapeutic payloads results in prolonged tumor retention, enhanced efficacy, and reduced systemic toxicity.

In this work, we developed a locally retained collagen-anchored α4-1BB agonist, termed α4-1BB-LAIR, by fusing an α4-1BB agonist to the ectodomain of an endogenous collagen binding protein, Leukocyte Associated Immunoglobulin Like Receptor 1 (LAIR1). Tested in combination with an antitumor antibody, TA99, in a fully syngeneic and poorly immunogenic B16F10 murine melanoma model, this combination exhibited little efficacy. Intriguingly, depletion of CD4+ T cells led to long-term durable cures in >90% of TA99- + α4-1BB-LAIR-treated animals. However, nearly all of these mice were unable to control a secondary tumor rechallenge. We hypothesized that the depletion of Tregs, which comprise a subset of CD4+ T cells, was driving this synergy. Tregs are immunosuppressive CD4+ T cells that express the transcription factor forkhead box protein P3 (Foxp3) and are critical to maintaining homeostasis and preventing autoimmunity[37,38]. Indeed, Foxp3−/− mice die at a young age from severe lymphoproliferative disease, systemic depletion of Tregs in adult mice leads to rapid lethal autoimmunity, and FOXP3 mutations in humans cause severe immune dysregulation[39–42]. Although Tregs play a critical role in curbing autoreactive T cells, they also constrict productive antitumor immune responses through a variety of mechanisms and at various stages of the tumor-immunity cycle[43,44].

Using flow cytometry and bulk-RNA sequencing, we probe the immunological mechanism of this synergy and find that CD4+ T cell depletion leads to an enhanced activation state in the tumor draining lymph node (TdLN), generating an influx of newly primed CD8+ T cells into the tumor. Local remodeling of the tumor microenvironment by TA99 and α4-1BB-LAIR enhances the cytotoxicity of these newly primed T cells, leading to tumor cell killing and eventual complete tumor regression. Using a Foxp3-DTR mouse model, which allows for selective depletion of Tregs only, we confirm that Treg depletion alone is sufficient for this synergy. Finally, we demonstrate that CD4+ T cell depletion can be replaced with a more clinically relevant agent known to enhance CD8+ T cell priming, αCTLA-4, without compromising efficacy. This combination of TA99 + α4-1BB-LAIR + αCTLA-4 also results in the formation of robust immunological memory, enabling rejection of a secondary tumor rechallenge. This work suggests that locally retained 4-1BB agonist and antitumor antibody therapy can be highly efficacious when combined with modalities that enhance T cell priming, which can be restrained by TdLN Tregs. Furthermore, this work supports the development of therapeutic strategies that specifically deplete and/or inhibit TdLN Tregs.

## Results

### TA99 + α4-1BB-LAIR synergizes robustly with CD4 compartment depletion

In order to develop a tumor-localized 4-1BB agonist, we leveraged a collagen anchoring strategy previously validated by our lab and others. We recombinantly expressed an α4-1BB antibody with previously demonstrated in vivo agonistic activity (clone LOB12.3,

Table S1) as a C-terminal fusion with the ectodomain of murine LAIR1, an endogenous immune cell inhibitory receptor that naturally binds collagen[45–48]. We verified that α4-1BB-LAIR expresses without aggregation (Fig. S1A), is able to bind plate bound collagen I via ELISA (Fig. S1B), and that binding to cell-surface expressed 4-1BB and the agonistic activity of the antibody fusion are minimally affected by the LAIR fusion (Fig. S1C–E).

To validate intratumoral retention without confounding target-mediated drug disposition (TMDD) effects, we also generated an αFITC-LAIR control antibody as this antibody has no known murine target (Table S1). Fluorescently labeled αFITC-LAIR administered intratumorally was preferentially retained in the tumor over unanchored αFITC antibody when measured longitudinally with IVIS (Fig. S1F, G).

α4-1BB agonist Urelumab is being clinically tested in combination with antitumor antibodies Rituximab, Cetuximab, and Elotuzumab which target CD20, EGFR, and SLAMF7, respectively (NCT01775631, NCT02110082, NCT02252263). Preliminary data has not been encouraging, with early reports from the Rituximab combination suggesting that Rituximab + Urelumab is no more efficacious than Rituximab monotherapy[49]. We sought to evaluate if collagen-anchoring would improve the efficacy of combination α4-1BB agonist + antitumor antibody therapy. Mice were inoculated with B16F10 melanoma flank tumors and treated systemically (intraperitoneally, or i.p.) with TA99, an antitumor antibody that binds to Trp1 expressed on the surface of B16F10 cells, followed by intratumorally (i.t.) administered α4-1BB-LAIR one day later for a total of 4 weekly cycles (This combination of TA99 + α4-1BB-LAIR is referred to collectively as the treatment, or "Tx", henceforth, Fig. 1a). Although this combination led to a statistically significant growth delay compared to PBS treated mice, nearly all mice eventually succumbed to their tumor burden, with only ~5% of mice achieving a complete response (CR, defined as no palpable tumor at day 100) (Fig. 1b). Notably, this growth delay was only slightly better than the individual components of Tx, although it was the only therapy with any complete responders (Fig. S2A).

In an effort to improve this combination therapy, we explored which cell types were critical for response. Surprisingly, we observed that when we also treated these mice with an αCD4 antibody that depletes the entire CD4+ T cell compartment, the complete response rate of TA99 + α4-1BB-LAIR improved dramatically, with >90% of mice achieving a complete response (Fig. 1b). However, when long-term survivors were rechallenged on the contralateral flank >100 days after initial tumor inoculation, nearly all mice succumbed to these secondary tumors (Fig. 1c). This was indicative of the inability of these mice to develop robust immune memory to B16F10 tumor cells, likely resulting from the depletion of CD4+ effector T cells which are necessary for proper formation of CD8+ memory T cells[50,51].

Growth delay with systemically administered αCD4 and α4-1BB has been reported previously, but we find that our specific components were necessary to achieve maximum efficacy, including TA99 (P = 0.0032) and, notably, retention via collagen anchoring (P = 0.0289) (Fig. 1d)[52]. Consistent with other preclinical reports with this α4-1BB antibody clone, no signs of toxicity were observed for the full therapeutic combination with or without collagen anchoring (Fig. S2B)[48]. Using 2.5F-Fc, an integrin-targeting antibody-like therapy, as an antitumor antibody equivalent, we also observed that this treatment paradigm is also efficacious in the MC38 colon carcinoma tumor model, albeit with a less dramatic complete response rate (Fig. S3)[53]. Although αCD4 drastically improved the efficacy of Tx, the lack of immune memory formation and low translational potential of long term αCD4 treatment motivated us to understand the mechanism of this synergy and ultimately develop alternative clinically relevant synergistic combinations.

### αCD4 improves priming in the TdLN

We investigated the chemokine/cytokine profile of the TME following treatments with PBS, Tx, Tx + αCD4, or αCD4 both 3 and 6 days after α4-1BB-LAIR administration. We dissociated tumors and analyzed the cytokine and chemokine milieu using a multiplexed flow cytometry-based ELISA assay. Although we observed general increases in inflammatory cytokines and chemokines in all treatment groups, only GM-CSF was specifically upregulated in the Tx + αCD4 group when compared to Tx or αCD4 alone (Fig. S4A). However, neutralization of this cytokine did not abrogate therapeutic efficacy of Tx + αCD4,

indicating that this spike in GM-CSF was dispensable for therapeutic efficacy (Fig. S4B).

We then used flow cytometry to analyze the tumors and tumor-draining lymph nodes (TdLNs) of mice treated with Tx, Tx + αCD4, or αCD4 again 3 and 6 days after the first α4-1BB-LAIR treatment. As expected, we observed complete depletion of total CD4⁺ T cells and Tregs (defined as Foxp3⁺ CD25⁺ CD4⁺ T cells) in the tumor (Fig. 2a) and TdLN (Fig. 2b) in both the Tx + αCD4 and the αCD4 groups.

Using CD44 and CD62L gating, we divided CD8⁺ T cells in the TdLN into naïve (CD44⁻ CD62L⁺), effector/effector memory (CD44⁺

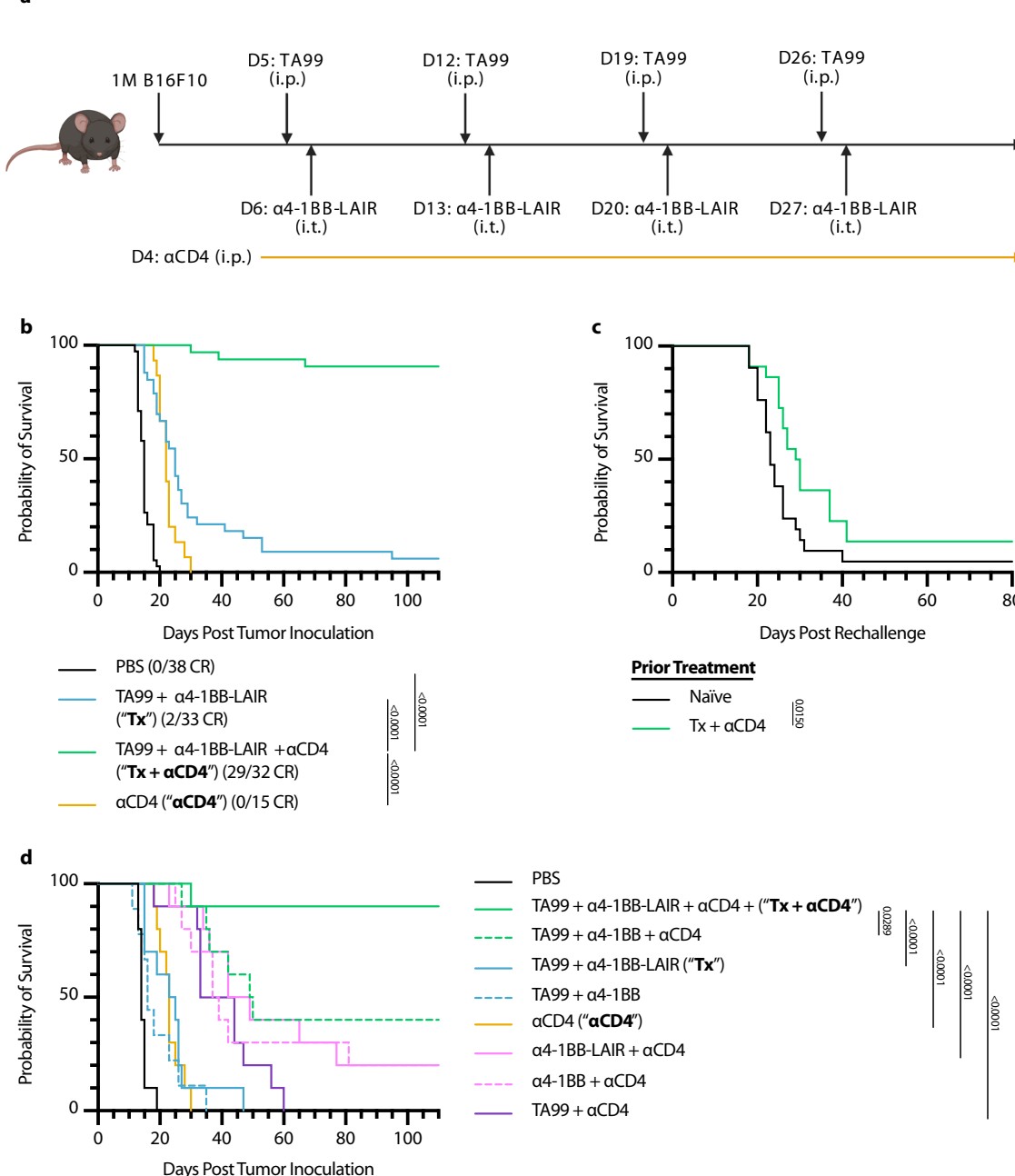

**Fig. 1 | TA99 + α4-1BB-LAIR synergizes robustly with CD4⁺ T cell depletion.** Mice were inoculated with 1 × 10⁶ B16F10 cells on day 0. **a** Treatment schedule of TA99 + α4-1BB-LAIR + αCD4. Mice were treated with 200 µg of TA99 (i.p.) on days 5, 12, 19, and 26, treated with 36.1 µg α4-1BB-LAIR (i.t.) on days 6, 13, 20, and 27 (molar equivalent to 30 µg α4-1BB), and treated with 400 µg αCD4 (i.p.) every 3 days starting 1 day before the first dose of TA99 and ending one week after the last dose of α4-1BB-LAIR (days 4 to 34). **b** Aggregate survival of mice treated with PBS

(n = 38), TA99 + α4-1BB-LAIR ("**Tx**") (n = 33), TA99 + α4-1BB-LAIR + αCD4 ("**Tx + αCD4**") (n = 32), or αCD4 (n = 15) (eight independent studies). **c** Survival of complete responders to Tx + αCD4 re-challenged on the contralateral flank >100 days after primary tumor inoculation. **d** Overall survival of mice treated with indicated combination variants, demonstrating all components are necessary for maximum efficacy (n = 9-10, two independent experiments). Survival was compared using log-rank Mantel-Cox test. "n.s." = not significant (P > 0.05).

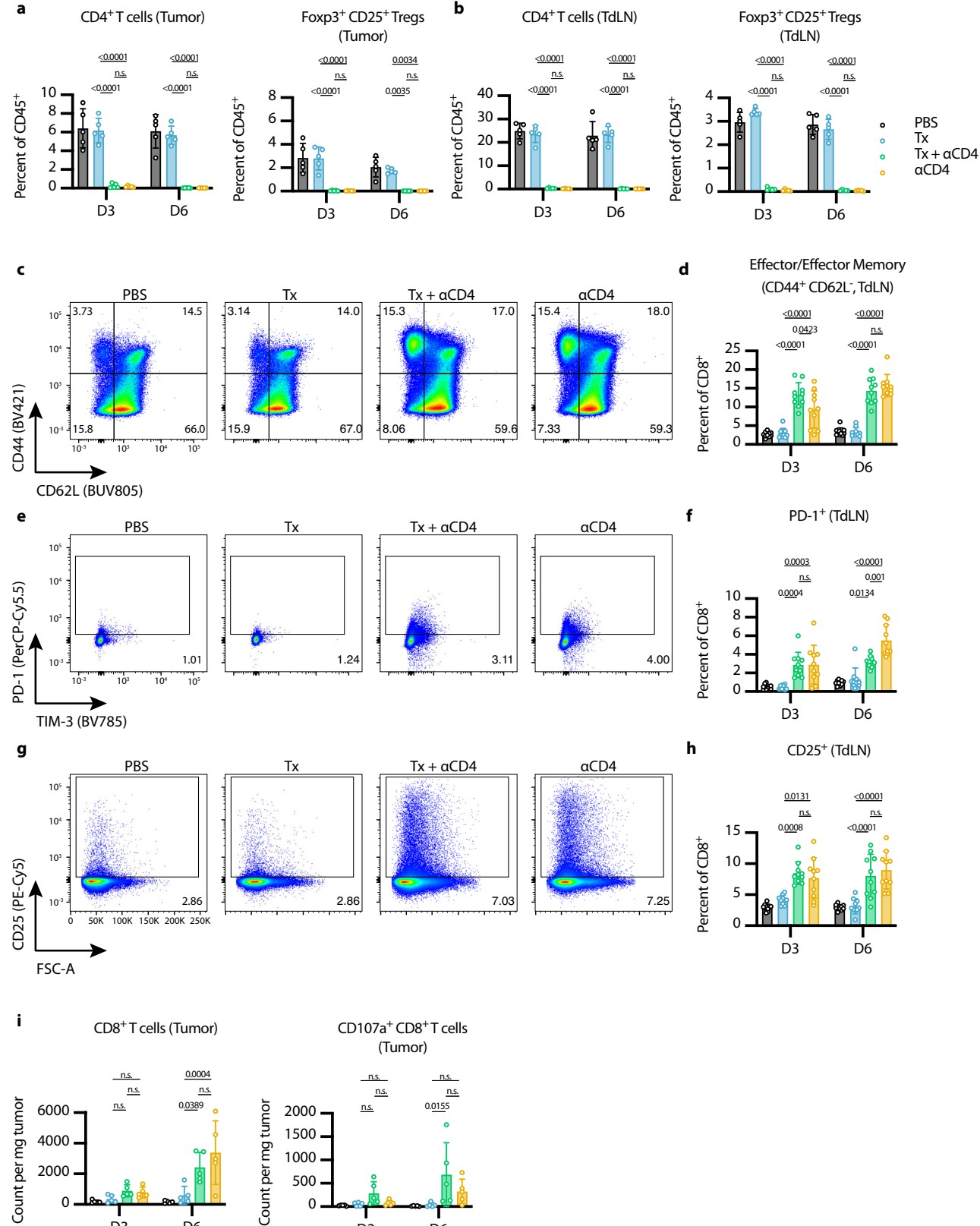

CD62L⁻), and central memory (CD44⁺ CD62L⁺) phenotypes. At both time points, we observed a shift of the CD8⁺ T cell population towards an effector/effector memory phenotype in the Tx + αCD4 and αCD4 groups (Fig. 2c, d). Additionally, we observed increases in both PD-1⁺ CD8⁺ T cells (Fig. 2e, f) and CD25⁺ CD8⁺ T cells (Fig. 2g, h), at both time points in the Tx + αCD4 and αCD4 groups, both of which are markers of

recently activated CD8⁺ T cells in lymphoid tissue. The magnitude of these changes was equivalent between the Tx + αCD4 and αCD4 groups, indicating that the αCD4 antibody component was driving these changes to the TdLN. There were largely no differences in the balance between stem-like (TCF1⁺ TIM-3⁺) vs. terminally differentiated (TCF1⁻ TIM3⁺) cells in the CD8⁺ PD-1⁺ T cell compartment among the

**Fig. 2 | αCD4 leads to new wave of CD8⁺ T cell priming, and Tx supports cytotoxicity of these cells in the tumor. a, b** Flow cytometry quantification (mean ± SD) of CD4⁺ T cells (gated on single cell/live/CD45⁺/CD3⁺NK1.1⁻/CD4⁺) and Tregs (gated on single cell/live/CD45⁺/CD3⁺NK1.1⁻/CD4⁺/Foxp3⁺CD25⁺) in the (**a**) tumor and (**b**) TdLN 3 and 6 days after first α4-1BB-LAIR treatment (*n* = 5). **c** Representative gating of CD44 and CD62L to define effector/effector memory CD8⁺ T cells in TdLN 6 days after first α4-1BB-LAIR treatment and (**d**), quantification (mean ± SD) of these cell populations 3 and 6 days after first α4-1BB-LAIR treatment (gated on single cell/live/CD45⁺/CD3⁺NK1.1⁻/CD8⁺/CD44⁺CD62L⁻, *n* = 10, two independent experiments). **e** Representative gating of PD-1⁺ CD8⁺ T cells 6 days after first α4-1BB-LAIR treatment and (**f**), quantification (mean ± SD) of these cell populations 3 and 6 days after first α4-1BB-LAIR treatment (gated on single cell/live/CD45⁺/CD3⁺NK1.1⁻/CD8⁺/PD-1⁺, *n* = 10, two independent experiments). **g** Representative gating of CD25⁺ CD8⁺ T cells 6 days after first α4-1BB-LAIR treatment and (**h**), quantification (mean ± SD) of these cell populations 3 and 6 days after first α4-1BB-LAIR treatment (gated on single cell/live/CD45⁺/CD3⁺NK1.1⁻/CD8⁺/CD25⁺, *n* = 10, two independent experiments). **i** Flow cytometry quantification (mean ± SD) of CD8⁺ T cells (left) and CD107a⁺ CD8⁺ T cells (right) (gated on single cell/live/CD45⁺/CD3⁺NK1.1⁻/CD8⁺) in the tumor 3 and 6 days after first α4-1BB-LAIR treatment (*n* = 5). Flow cytometry data was compared using two-way ANOVA with Tukey's multiple hypothesis testing correction. "n.s." = not significant (*P* > 0.05).

various treatment groups, although we observed a trend towards less terminally differentiated cells in the tumor in the Tx + αCD4 group (Fig. S5).

Six days following treatment with either Tx + αCD4 or αCD4, we observed increased CD8⁺ T cells infiltrating the tumor (Fig. 2i), which is in agreement with the enhanced activation state observed in the TdLN (Fig. 2e–h). This result is consistent with previous preclinical and clinical studies that have shown treatment with αCD4 can enhance T cell priming, leading to increased numbers of tumor reactive CD8⁺ T cells[54–56]. However, only in the Tx + αCD4 group, when compared to PBS or Tx alone, do we observe an increase in degranulating CD107a⁺ CD8⁺ cytotoxic T cells (Fig. 2i, Fig. S6A). No major differences in 4-1BB expression on CD8+ T cells were detected in the tumor and only minor increases were seen on CD8⁺ T cells in the TdLN in Tx + αCD4 and αCD4 treated mice (Fig. S6B–D). These data suggest that αCD4 therapy, independent of Tx, induces de novo priming in the TdLN, leading to more CD8⁺ T cell infiltration in the tumor. However, we hypothesized that Tx supports these newly primed cells and maintains their cytotoxic phenotype within the tumor, leading to eventual tumor regression.

## TdLN has increased proliferation and T cell gene signatures by Bulk-RNA sequencing

To further interrogate immunological changes to the TdLN and tumor in an unbiased holistic manner, we performed bulk RNA-sequencing on CD45⁺ cells from TdLN samples from mice treated with PBS, Tx, Tx + αCD4, or αCD4 3 and 6 days following α4-1BB-LAIR administration. We generated a UMAP plot of the TdLN samples and found that, at the bulk transcript level, large differences between samples were apparent only at the later time point (Fig. 3a). Additionally, sample clustering at this later time point was driven entirely by αCD4, with the αCD4 and Tx + αCD4 samples clustering separately from the PBS and Tx samples. In fact, we observed almost no differentially expressed genes (DEG) in the TdLN when comparing Tx + αCD4 versus αCD4 or Tx versus PBS-treated samples (Fig. 3b), indicating that Tx alone had no appreciable change on the transcriptional program in the TdLN.

To assess what changes αCD4 drove in the TdLN, we examined DEGs between Tx + αCD4 and Tx treated samples (Fig. 3b). We found 247 upregulated genes and 82 downregulated genes (FDR ≤ 5%, Fig. 3c). We used enrichR to determine which pathways these upregulated DEGs were enriched in[57–59]. Upregulated genes belonged to pathways involving cell cycling, DNA replication, and MYC related genes, indicative of a highly proliferative state in the TdLN. They were also enriched for both cycling and CD8+ T cell states (Fig. 3d). Overall, the TdLN transcriptional data demonstrated that 1) changes to the TdLN resulted from αCD4 treatment, independent of Tx, and 2) these changes led to enhanced proliferation and T cell activation in the TdLN.

## Tx + αCD4 leads to cytotoxic CD8⁺ T cell program in the tumor

We similarly used bulk-RNA sequencing to examine immune cell gene expression programs within the tumor. We performed hierarchical clustering of the tumor samples while also independently clustering all significant DEGs (with a log 2-fold change ≥2 or ≤−2 and p-adj ≤ 0.05) using k-means clustering. This clustering identified 10 distinct gene clusters of co-expressed genes. Samples clustered imperfectly by treatment type, with two of the three Tx + αCD4 day 6 samples showing distinct transcriptional programs (Fig. 4a, b, Fig. S7). These two samples had the smallest tumor size at time of necropsy, indicating they were already robustly responding to therapy at this time point. We next performed pathway enrichment analysis on the individual gene clusters. Of particular interest were clusters 1 and 2 (and to a lesser extent cluster 4), which were upregulated specifically in the Tx + αCD4 groups, and cluster 3, which contains genes upregulated in both the Tx + αCD4 and Tx groups and represents a Tx-specific transcriptional program (Fig. 4b). These clusters are enriched for a range of GO terms associated with productive cellular immune responses (regulation of T cell activation, alpha-beta T cell activation, lymphocyte-mediated immunity, etc.). However, only clusters 1 and 2 were enriched for genes associated with interferon-gamma (IFNγ) production, suggesting that Tx alone is not sufficient to drive IFNγ production (Fig. 4c). Notably, because Tx + αCD4 and αCD4 drive similar levels of increased CD8+ T cell counts (Fig. 2f), but cytotoxic genes are only enriched in Tx + αCD4, we can conclude that this cytotoxic signature is not an artifact of increased CD8+ T cell counts. Cluster 7, which is highly expressed in PBS samples, contained genes enriched for, among others, pigmentation gene programs, likely representing increased CD45⁻ tumor cells in this sample (Fig. 4c).

To further assess changes to the tumor microenvironment, we looked at DEGs between Tx + αCD4 tumor samples 3 and 6 days following α4-1BB-LAIR. 63 genes were upregulated and 43 genes downregulated between these two time points (FDR ≤ 5%, Fig. 5a). We used the upregulated DEGs to establish a "response" signature for Tx + αCD4. We then asked if this gene signature was expressed in any other treatment conditions/time points. Indeed, this signature was highly expressed only in the Tx + αCD4 late time point, indicating this was a bona fide response signature unique to Tx + αCD4 treated mice (Fig. 5b). We then performed pathway enrichment analysis to determine what pathways these genes were associated with. Confirming our previous flow data, we saw effector and effector memory T cell signatures. Additionally, we saw genes associated with TCR signaling, interleukin-2 (IL-2) signaling and STAT5A activity (Fig. 5c). Recent literature has highlighted a role for IL-2, or more broadly STAT5Aa activity, in amplifying T cell populations that drive responses to checkpoint blockade[60–62]. Gene-set enrichment analysis (GSEA) revealed that while both Tx + αCD4 and αCD4 drove an enrichment of a memory precursor effector cell (MPEC) signature, only Tx + αCD4 drove a corresponding statistically significant de-enrichment of a short-lived effector cell (SLEC) signature and a robust enrichment of CD8⁺ T cell signature associated with cell survival, activation, memory, and response to checkpoint blockade therapy (Fig. 5d). Furthermore, GSEA revealed that while Tx or αCD4 individually drove an enrichment of a CD8⁺ T cell exhaustion signature, Tx + αCD4 did not. Taken together, the tumor transcriptional data support the notion that Tx + αCD4 drives a robust cytotoxic T cell program leading to tumor rejection.

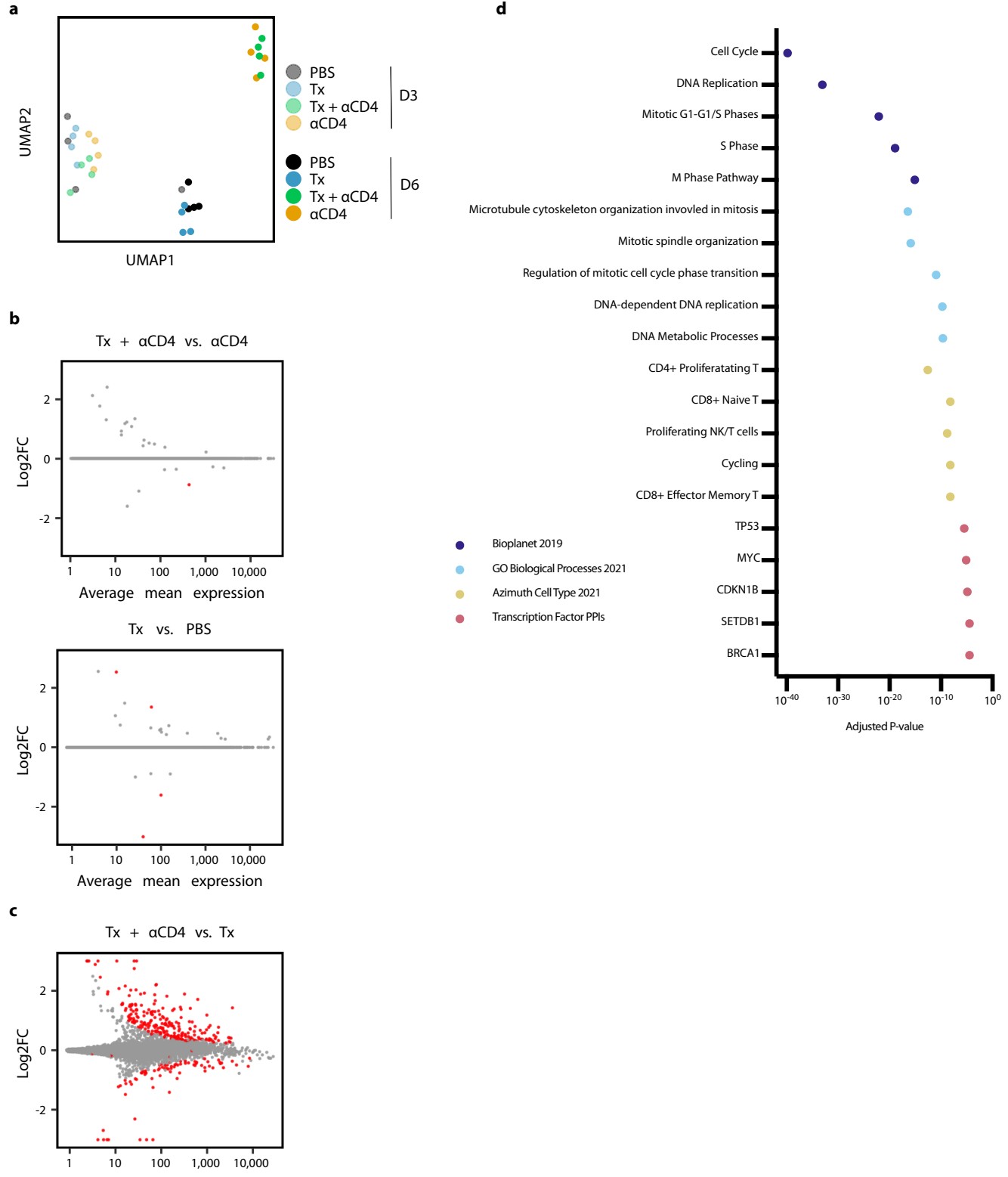

**Fig. 3 | αCD4 drives proliferation in the TdLN. a** UMAP plot of TdLN transcriptomes (*n* = 4 per group). **b** Differential expression testing of Tx + αCD4 vs. αCD4 and Tx vs. PBS TdLN samples 6 days after first α4-1BB-LAIR treatment, with statistically significant hits highlighted in red (FDR ≤ 5%). **c** Differential expression testing of Tx + αCD4 vs. Tx TdLN samples 6 days after to first α4-1BB-LAIR treatment, with statistically significant hits highlighted in red (FDR ≤ 5%). **d** Pathway enrichment analysis of upregulated DEGs identified in (**c**). Pathway enrichment analysis was performed using Fisher's exact test with Benjamini-Hochberg multiple hypothesis testing correction.

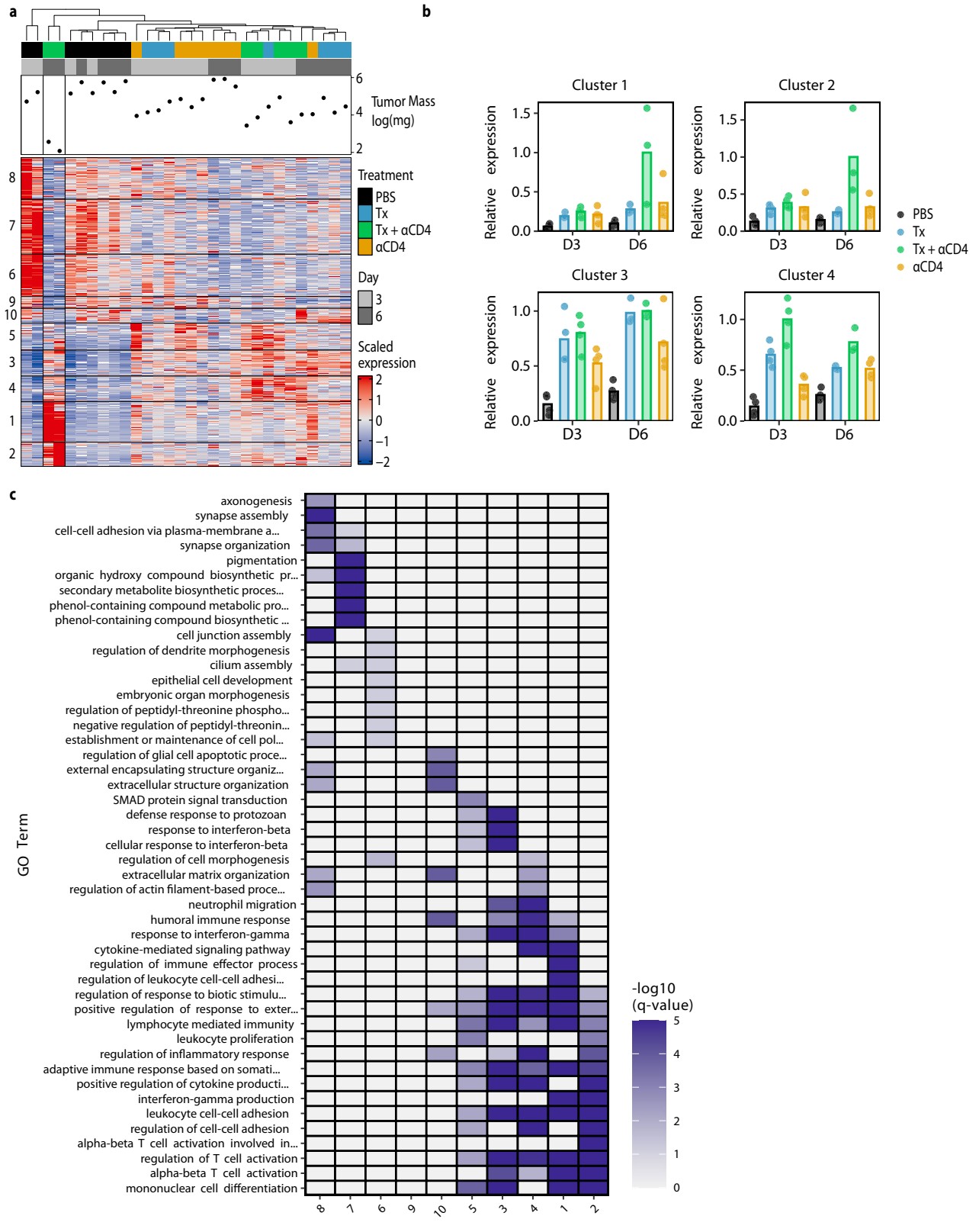

**Fig. 4 | Tx + αCD4 upregulated gene clusters enriched for CD8⁺ effector programs. a** Heatmap of *k*-means clustered DEGs (absolute value lfc ≥ 2, FDR ≤ 10%) and tumor samples hierarchically clustered (*n* = 3–4). **b** Normalized expression of select individual gene clusters identified in (**a**), for each experimental condition. **c** Pathway enrichment analysis for each gene cluster identified in (**a**).

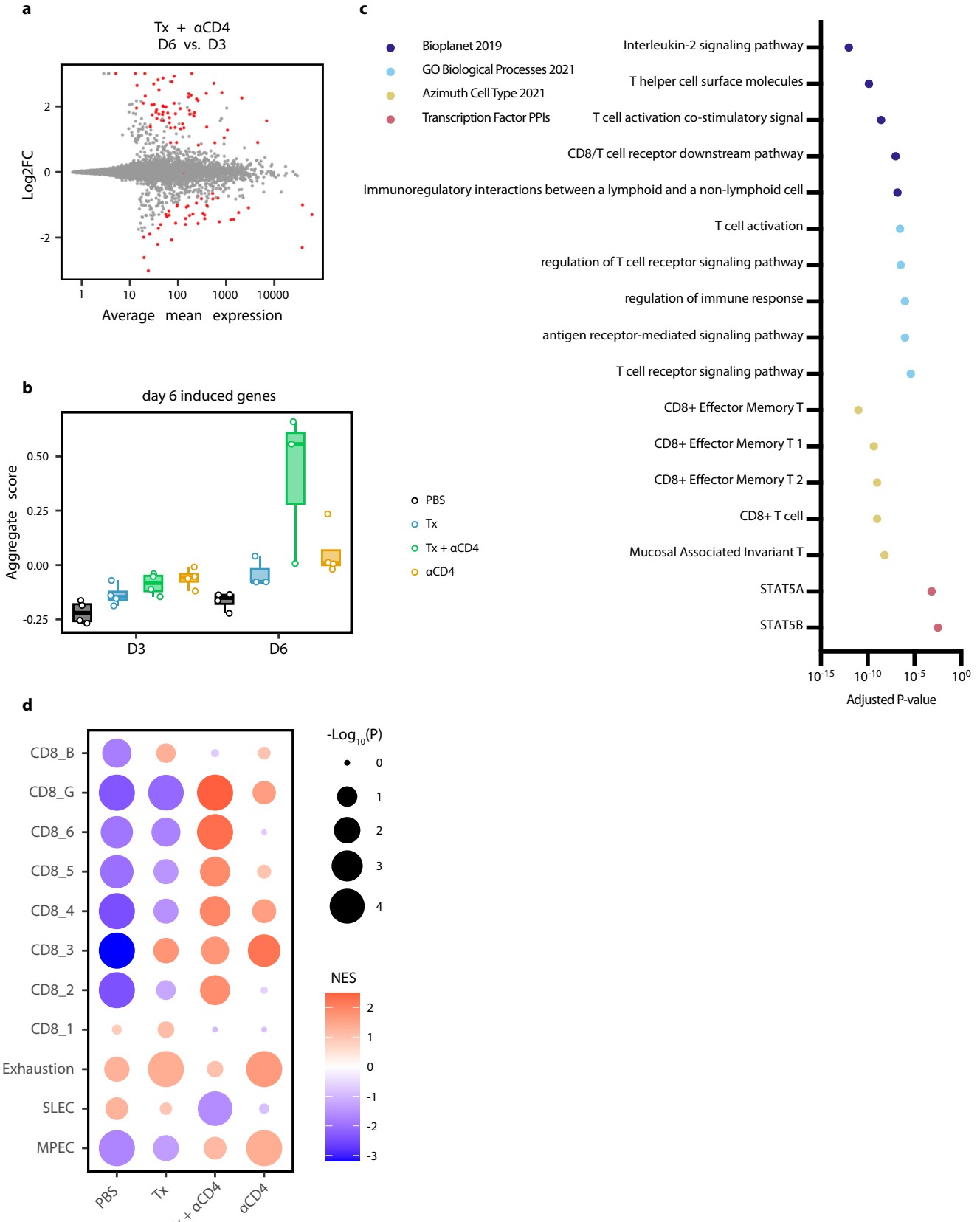

## Treg depletion results in equivalent efficacy as whole CD4 compartment depletion

We hypothesized that Treg depletion was the primary functional consequence of αCD4 therapy, and that Treg-specific elimination would lead to similar efficacy in combination with Tx. To test this hypothesis, we turned to Foxp3-DTR mice, which express the diphtheria toxin receptor (DTR) and GFP under the control of the Foxp3 promoter. In these mice, all Foxp3+ cells are also DTR+, and thus

**Fig. 5 | Tx + αCD4 associated with cytotoxic T cell signature in the tumor.**
**a** Differential expression testing of Tx + αCD4 on day 3 vs. day 6 tumor samples relative to first α4-1BB-LAIR treatment, with statistically significant hits highlighted in red (FDR ≤ 5%). **b** Average expression level of significantly upregulated DEGs identified in (**a**) across all treatment groups (n = 3–4). **c** Pathway enrichment analysis of upregulated DEGs identified in (**a**). **d** Gene-set enrichment analysis (GSEA) of day 6 tumor samples. Indicated treatment group is compared against all other treatment groups to determine enrichment of listed gene sets. NES = normalized enrichment score. Pathway enrichment analysis was performed using Fisher's exact test with Benjamini–Hochberg multiple hypothesis testing correction. The boxplot extends from the 25th percentile to the 75th percentile, with the center line representing the median value. The whiskers extend up to 1.5 times the interquartile range (difference between the 75th and 25th percentile) from the edge of the box, not exceeding the further datapoint.

susceptible to diphtheria toxin (DT) mediated cell death. Systemic administration of DT to these mice leads to rapid and complete depletion of nearly all Foxp3$^+$ Tregs. However, with repeat dosing these mice succumb to lethal autoimmunity within 10-20 days of DT administration[39]. In order to facilitate long-term depletion of Tregs in the tumor and TdLN without inducing lethal autoimmunity, we developed a low dose, intratumoral diphtheria regimen. Every other day intratumoral dosing of 75 ng or 125 ng of DT depleted tumor and TdLN to similar levels as 1 μg of systemically administered DT, with reduced impacts on splenic Treg populations (Fig. 6a, Fig. S8A). Additionally, we did not observe signs of toxicity, as measured by weight loss, with intratumoral low dose DT, while mice receiving systemic DT showed trends of weight loss at time of euthanasia (Fig. 6b, Fig. S8B). Thus, we felt confident that low-dose intratumoral DT was a safe and effective model system to achieve long-term intratumoral and intranodal Treg depletion

B16F10 tumor-bearing Foxp3-DTR mice were treated with Tx + αCD4, Tx + DT, or DT alone. To allow for lesions of sufficient size for intratumoral administration of DT, the absolute timing of therapy administration was delayed two days for all groups (such that DT and αCD4 were initiated on day 6). Mice receiving Tx + DT responded equally as well as mice receiving Tx + αCD4, with a trend (but not statistically significant) towards a higher complete response rate in the Tx + DT group (Fig. 6c). Interestingly, DT on its own also resulted in significant growth delay, but ultimately almost all mice succumbed to their tumor burden. To confirm that the effect of DT was purely a result of Treg depletion, we treated WT mice with DT, which resulted in no different growth kinetics over PBS-treated mice. No signs of toxicity, as assessed by weight loss, were observed throughout the course of treatment (Fig. S8C). A previously published study demonstrated that transient DT given with systemic α4-1BB agonist therapy led to severe immune-related adverse events (irAE) in MC38 tumor-bearing mice, further highlighting the advantages of our collagen-anchored α4-1BB agonists[63]. Notably, when cured mice were rechallenged >100 days after their primary tumor inoculation, the majority of the Tx + αCD4 mice cured did not reject rechallenge, consistent with previous results, while 100% of mice cured with Tx + DT rejected this rechallenge, demonstrating that these mice had developed robust immunological memory against B16F10 tumor antigens (Fig. 6d). This result demonstrated that 1) elimination of Tregs is sufficient to boost the efficacy of Tx and 2) elimination of Tregs alone while maintaining the CD4$^+$ effector population allows for the proper formation of long term immune memory, consistent with other prior reports in the literature[50,51].

**Therapy induced de novo priming is necessary for therapeutic efficacy**

Our data suggest that αCD4 mediates an increase in CD8$^+$ T cell priming in the TdLN, which then leads to the accumulation of newly primed CD8$^+$ T cells in the tumor. However, an alternative explanation is that endogenous T cells already in the tumor locally proliferate and expand after αCD4 treatment. To test this hypothesis and assess if this intratumoral T cell expansion is critical to therapeutic efficacy, we treated tumor-bearing mice with FTY720 concurrent with Tx + αCD4. FTY720 is a small molecule S1PR antagonist that prevents lymphocyte egress from lymphoid tissues, thus blocking any contributions from therapy-

induced de novo priming to efficacy[64]. FTY720 was initiated concurrently with the start of αCD4 treatment. To give sufficient time for the endogenous T cell response to develop before FTY720 initiation, treatment initiation was delayed two days (such that αCD4 and FTY720 were initiated on day 6 following tumor inoculation).

The addition of FTY720 to Tx + αCD4 abrogated therapeutic efficacy, with no complete responders and only minor tumor growth delay in this treatment cohort (Fig. 6e). Indeed, when we examined the tumor compartment via flow cytometry, the addition of FTY720 to Tx + αCD4 dropped CD8$^+$ T cell counts back to baseline (PBS/DMSO) levels (Fig. 6f). This confirmed that increases in CD8$^+$ T cells in the tumor after Tx + αCD4 were due to de novo priming and trafficking from the TdLN and not local proliferation of T cells already in the tumor. The increased activation and proliferation in the TdLN (as measured by increased Ki67$^+$ CD8$^+$ T cells, increased CD25$^+$ CD8$^+$ T cells, and a shift to an effector/effector memory phenotype in the CD8$^+$ T cell population) was preserved with the addition of FTY720, confirming that FTY720 prevented trafficking of these newly primed T cells to the tumor (Fig. 6g, Fig. S9A). Indeed, beginning αCD4 therapy 8 days before tumor inoculation maintained some efficacy of the combination; however, delaying initiation of αCD4 therapy to day 10 abrogated efficacy, consistent with αCD4's role in priming (Fig. S9B).

Interestingly, if initiation of FTY720 therapy was delayed just two days (concurrent with α4-1BB-LAIR), therapeutic efficacy of this combination was restored and T cell counts in the tumor were restored to the same levels as Tx + αCD4 (Fig. S9C–E). For all FTY720 dosing schemes, blood T cell levels were significantly reduced compared to untreated mice, confirming that FTY720 was functioning as expected after treatment initiation (Fig. S9F). These data suggest that only a single priming wave is sufficient for the efficacy of Tx + αCD4, and this priming wave occurs in a narrow time frame of two days following αCD4 initiation.

**αCTLA-4 therapy also synergizes with TA99 + α4-1BB-LAIR**

Based on the presented data, we concluded that αCD4 synergizes with Tx by initiating a wave of de novo priming, that these new tumor-infiltrating T cells are supported by the local α4-1BB-LAIR agonist and TA99, and that this two-step process ultimately drives therapeutic efficacy. We therefore hypothesized that other modalities capable of improving priming, such as αCTLA-4, would also synergize well with TA99 + α4-1BB-LAIR. Although the dominant mechanism of αCTLA-4 is contested, the literature supports that treatment with αCTLA-4 improves T cell priming and infiltration into the tumor and increased TdLN exposure of αCTLA-4 has been shown to improve therapeutic outcomes[65–67]. We therefore treated B16F10-bearing mice with Tx + αCTLA-4, and found that this combination was also highly efficacious, with an ~80% complete response rate (Fig. 7a). We hypothesized that mice cured with Tx + αCTLA-4 would also generate robust immune memory and reject rechallenge as their CD4$^+$ effector T cell pool remained intact, consistent with prior literature on the role of CD4$^+$ effector T cells in CD8$^+$ memory T cell formation[50,51]. In agreement with this hypothesis, 100% of survivors rechallenged >100 days after initial tumor inoculation rejected this secondary tumor rechallenge (Fig. 7b). Although synergy between systemic α4-1BB and αCTLA-4 has been reported in the MC38 model, it was not previously reported to be efficacious in a B16 melanoma model, again highlighting the

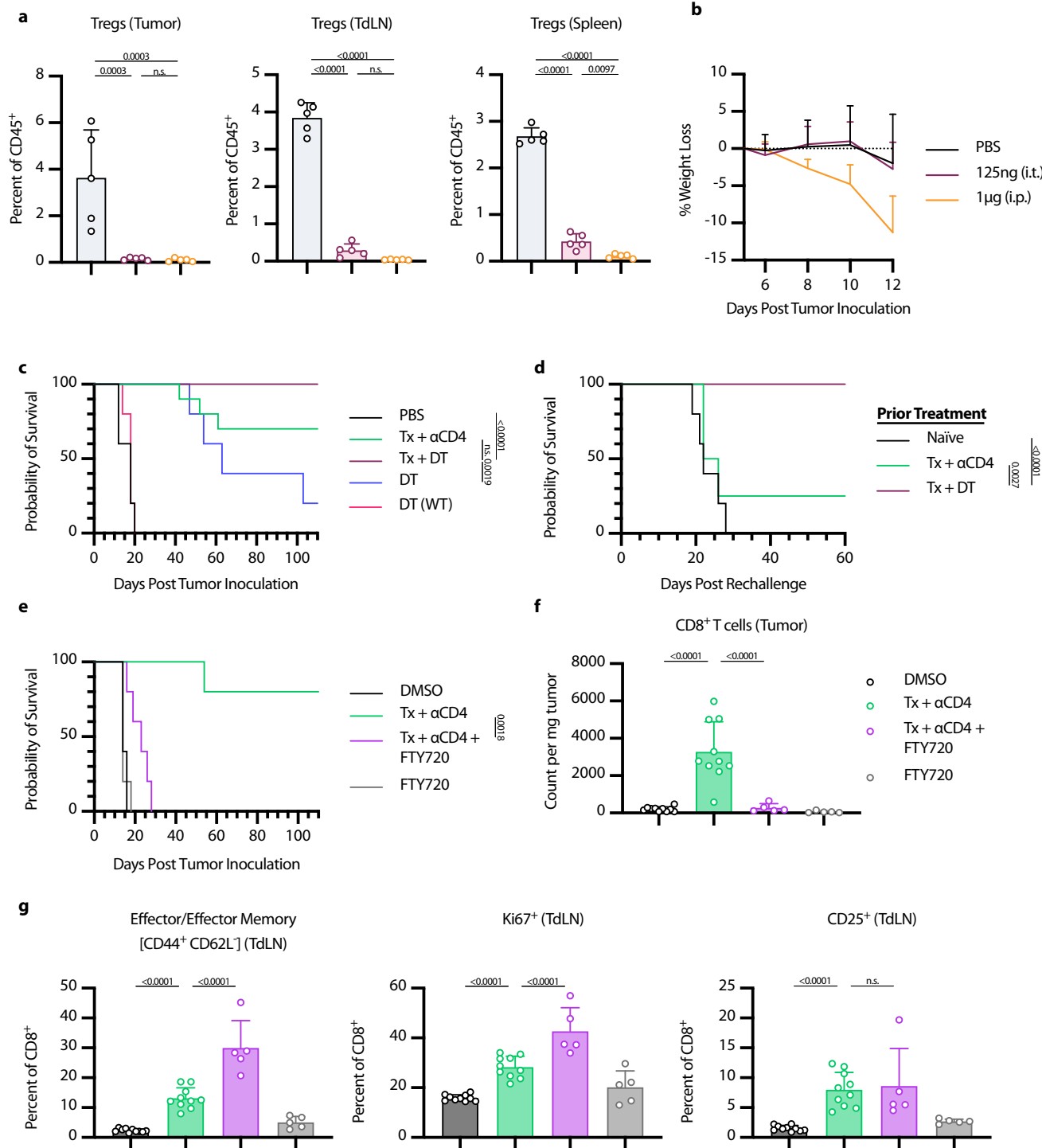

**Fig. 6 | Tx + αCD4 is Treg dependent and requires de novo priming for efficacy.**
**a–d** Foxp3-DTR Mice were inoculated with 1 × 10⁶ B16F10 cells on day 0. **a** Mice were treated on days 6, 8, and 10 with either 125 ng DT (i.t.) or 1 µg DT (i.p.). Flow cytometry quantification (mean ± SD) of Tregs in tumor, TdLN, or spleen on day 12 (gated on single cell/live/CD45⁺/CD3⁺NK1.1⁻/CD4⁺/GFP(Foxp3)⁺, n = 5). **b** Weight loss (mean+SD) of mice from (**a**). **c** Survival of Foxp3-DTR mice treated with PBS (n = 5), Tx + αCD4 (n = 10), Tx + DT (n = 10), DT (n = 5), WT mice treated with DT (n = 5). Mice were treated with the same relative dose/dose schedule as in Fig. 1a, but treatment initiation was delayed two days. DT-treated mice received 125 ng DT (i.t.) every other day from day 6 to day 36. **d** Survival of complete responders to Tx + αCD4 or Tx + DT re-challenged on the contralateral flank >100 days after primary tumor inoculation. **e–g** WT mice were inoculated with 1 × 10⁶ B16F10 cells on day 0. **e** Overall survival of mice treated with PBS/DMSO (n = 5), Tx + αCD4 (n = 5), Tx + αCD4 + FTY720 (n = 5), or FTY720 (n = 5). Mice were treated with the same relative

dose/dose schedule as in Fig. 1a, but treatment initiation was delayed two days. Mice were treated with 30 µg of FTY720 (i.p.) every other day from days 6 to 36. **f** Flow cytometry quantification (mean ± SD) of CD8⁺ T cell counts in tumor 6 days after first α4-1BB-LAIR treatment. (gated on single cell/Live/CD45⁺/CD3⁺NK1.1⁻/CD8⁺, n = 5 for Tx + αCD4 + FTY720 and FTY720 groups, n = 10 and two independent experiments for DMSO and Tx + αCD4 groups). **g** Flow cytometry quantification (mean ± SD) of effector/effector memory (CD44⁺ CD62L⁻), CD25⁺, and Ki67⁺ CD8⁺ T cells in the TdLN 6 days after α4-1BB-LAIR treatment (gated on single cell/live/CD45⁺/CD3⁺NK1.1⁻/CD8⁺, n = 5 for Tx + αCD4 + FTY720 and FTY720 groups, n = 10 and two independent experiments for DMSO and Tx + αCD4 groups). Flow cytometry data was compared using one-way ANOVA with Tukey's multiple hypothesis testing correction. Survival was compared using log-rank Mantel Cox test. "n.s." = not significant (P > 0.05).

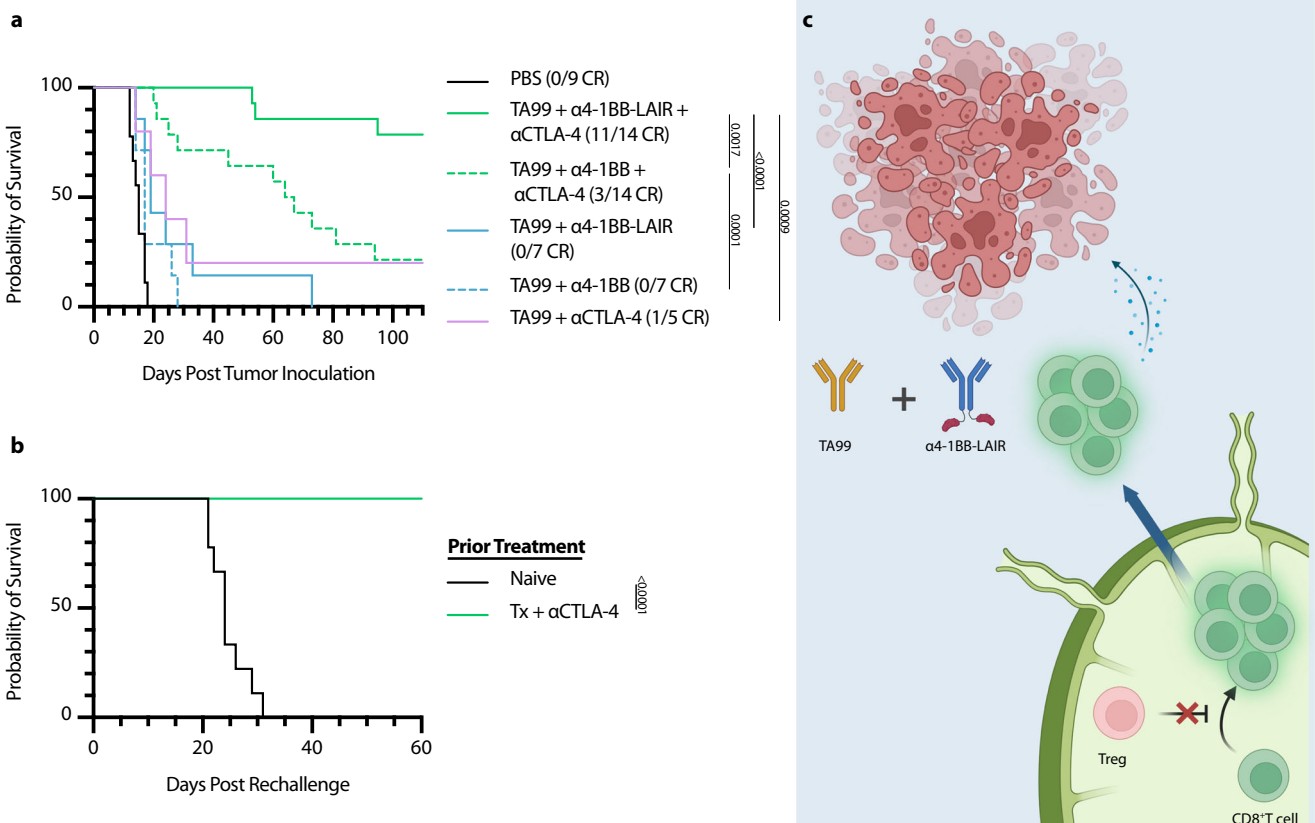

**Fig. 7 | αCTLA-4 can replace αCD4 while maintaining efficacy and rescuing memory formation.** Mice were inoculated with $1 \times 10^6$ B16F10 cells on day 0. **a** Overall survival of mice treated either with PBS ($n = 9$, two independent studies), TA99 + α4-1BB-LAIR + αCTLA-4 ($n = 14$, two independent studies), TA99 + α4-1BB + αCTLA-4 ($n = 14$, two independent studies), TA99 + α4-1BB-LAIR ($n = 7$), TA99 + α4-1BB ($n = 7$), or TA99 + αCTLA-4 ($n = 7$). Mice were treated with the same dose/dose schedule as in Fig. 1a with 200 µg αCTLA-4 (i.p.) given on days 6, 9, 13, 16, 20, 23, and 27. **b** Survival of complete responders to Tx + αCTLA-4 re-challenged on the contralateral flank >100 days after primary tumor inoculation. **c** Graphical abstract of proposed mechanism of action. Tregs in the TdLN constrain proper priming of tumor reactive CD8+ T cells, and inhibition or depletion of these cells results in a wave of newly primed CD8+ T cells entering the tumor, where their cytotoxic program is supported by TA99 and collagen-anchored α4-1BB-LAIR. Survival was compared using log-rank Mantel–Cox test. "n.s." = not significant ($P > 0.05$).

importance of the antitumor antibody and local retention through collagen anchoring in our therapy[68].

## Discussion

α4-1BB agonist antibodies have demonstrated robust efficacy as both a monotherapy and in combination with other immunotherapy agents in preclinical mouse models, but have so far failed in the clinic due to dose-limiting toxicities. In this work, we set out to develop α4-1BB antibodies with tumor-restricted activity via collagen anchoring. We have previously demonstrated that fusion of collagen-binding proteins lumican or LAIR to extended half-life versions of IL-2 and IL-12 improves efficacy and limits toxicities when directly injected into tumors, even in relatively collagen-sparse B16F10 melanoma tumors, such as those used in this study[27,28,36].

To generate collagen-anchored α4-1BB antibodies, we fused murine LAIR1 to the C-terminus of the heavy chain of an α4-1BB agonist antibody. We tested this agonist combination with a systemic antitumor antibody, TA99. We chose this combination because (1) α4-1BB agonist Urelumab is currently being tested in combination with antitumor antibodies Cetuximab, Rituximab, and Elotuzumab and (2) a wide range of other antitumor antibodies which recognize antigens expressed on tumor cells are currently in the clinic[69]. Antitumor antibodies have been demonstrated to improve antitumor immune responses by both generating antigenic cell debris to enhance T cell priming and also by reprogramming myeloid cells in the tumor through Fc:FcγR interactions[70]. In

agreement with preliminary phase 1 data, this combination did not result in robust efficacy in our hands, with only minor growth delay and complete responses in ~5% of treated mice[49]. However, we unexpectedly discovered that depletion of the entire CD4+ T cell compartment throughout the course of this combination therapy dramatically improved response rates, with >90% of mice achieving durable complete responses. A similarly efficacious combination (systemic α4-1BB + αCD4) has been reported in the literature, although durable responses were not seen, with all mice succumbing to their tumors between day 70-80[52]. To our knowledge, this is the highest complete response rate seen of any α4-1BB agonist antibody therapy in the poorly immunogenic B16F10 melanoma tumor model.

As Tregs comprise a sizable portion of the CD4+ T cell compartment, we tested Treg depletion in lieu of whole CD4+ T cell depletion using Foxp3-DTR mice in combination with TA99 + α4-1BB-LAIR and observed equivalent efficacy. While Tregs play a crucial role in preventing autoimmunity, they also constrain productive antitumor immune responses. Intratumoral Treg infiltration is correlated with poor prognosis across many different tumor types and there is evidence that intranodal Tregs infiltration is a better predictor of survival than blood or intratumoral Tregs in certain contexts[71–74]. Tregs exert their effects through multiple different pathways, including secretion of immunosuppressive cytokines such as IL-10, Transforming Growth Factor-beta (TGF-β), and IL-35, acting as a sink for IL-2 due to their high expression of the IL-2 high-affinity receptor CD25, generation of

immunosuppressive adenosine through expression of CD39, and expression of inhibitory receptors such as CTLA-4 and LAG-3[43,44].

Tregs are a major contributor to the immunosuppressive environment of the tumor, but they can also interfere with CD8+ T cell priming in lymphoid tissues[75,76]. Even prior to the identification of the transcription factor Foxp3 as the canonical driver of Tregs, seminal work found that depletion of CD25+ T cells (a subset of which are Tregs) before tumor implantation can lead to enhanced antitumor immune responses and eventual spontaneous tumor rejection[77]. Although how Tregs constrain priming is multifaceted, it is well established that CTLA-4 expressed on Tregs can transendocytose CD80 and CD86 off the surface of dendritic cells, hampering their ability to provide proper co-stimulation and prime CD8+ T cells[78–80]. Blocking this transendocytosis is thought to at least partially explain the mechanism of how αCTLA-4 therapy functions to improve priming. The αCTLA-4 antibody used in this study has also been shown deplete intratumoral Tregs, although peripheral Tregs (including those in TdLNs) instead expand with treatment[81,82]. Recent work from our own group has demonstrated that Treg depletion alone is not sufficient for the maximum efficacy of αCTLA-4 therapy, supporting the notion that enhanced priming is a critical mechanism of action of αCTLA-4 therapy[67]. Indeed, we show αCTLA-4 synergized as well as complete CD4+ T cell or Treg-specific depletion with TA99 + α4-1BB-LAIR. Our work supports the notion that intranodal Tregs dampen antitumor immune responses by constraining proper priming, and that relieving this constraint can bolster the magnitude of the antitumor T cell response and synergize robustly with T cell-directed agonist immunotherapies, particularly in immunologically cold tumors such as the one used in this study.

Although long-term CD4+ T cell compartment depletion leads to obvious defects in both T and B cell adaptive immune responses, transient CD4+ T cell depletion has been clinically tested in cancer and other disease states using an αCD4 antibody. Transient αCD4 depletion resulted in similar increases in de novo priming and CD8+ T cell infiltration in the tumor, consistent with our own data[55,56]. However, although no adverse events have been reported in these small phase 1 trials, these patients are still at risk of severe and possibly fatal infections if exposed to pathogens while devoid of their CD4+ compartment. Additionally, although αCD4 depletion therapy synergized well with TA99 + α4-1BB-LAIR, mice failed to form immunological memory, which can be important for long-term tumor control and control of distant metastases. It is well documented that CD4+ helper T cells are crucial for both enhanced priming of CD8+ T cells and proper formation of long-term T cell memory[50,51,83,84]. In patients, the presence of memory T cells corresponds with breast cancer survival and memory T cells have been reported to persist in survivors of melanoma treated with immunotherapy for at least 9 years[85,86]. With this in mind, we set out to understand the mechanism of how CD4+ T cell compartment depletion synergized with TA99 + α4-1BB-LAIR and develop new combination therapies with higher translational potential. Our data demonstrated that CD4+ T cell depletion eliminated Tregs in the TdLN, removing immunosuppressive constraints on proper CD8+ T cell priming, and induced a wave of freshly primed T cells to enter the TME. The combination of TA99 + α4-1BB-LAIR is able to reprogram the TME into a more supportive environment for these newly primed T cells, allowing them to maintain their cytotoxic phenotype, leading to tumor regression and clearance (Fig. 7c). Indeed, recent data has suggested a two-step model for CD8+ T cell activation in cancer, with initial activation in the TdLN and effector differentiation occurring with co-stimulation in the tumor[87]. The two components of our therapy mirror this paradigm, with αCD4 increasing activation in the TdLN and TA99 + α4-1BB-LAIR enhancing effector functions of these newly activated CD8+ T cells directly in the tumor. Although we have limited our analysis to the TdLN, both αCD4 and αCTLA-4 were given systemically and our local DT administration still elicited some level of splenic Treg

depletion, so we cannot rule out other lymphoid tissues, particularly the spleen, playing a role in the priming effects we have observed in these therapeutic combinations. Additionally, although we saw convincing transcriptional signatures in both the TdLN and tumor supporting our conclusions, our analysis was limited to bulk RNA-sequencing of all CD45+ cells. We are unable to ascertain specific changes in frequency to different cellular populations. A more granular approach, such as single-cell RNA-sequencing, would provide us with additional clarity on the mechanism of action of this triple combination therapy.

This localized therapy is reliant on intratumoral administration of the α4-1BB-LAIR component, which is clinically feasible with advances in interventional radiology[88–90]. Indeed, the oncolytic virus therapy talimogene laherparepvec (T-vec) has been approved since 2015 and is routinely injected into cutaneous and subcutaneous unresectable melanoma lesions[91,92]. Preclinical and clinical development around intratumorally administered therapies have been steadily on the rise. Notably, there is an ongoing clinical trial intratumorally administering α4-1BB agonist Urelumab (NCT03792724), demonstrating that this is a feasible approach for an antibody therapeutic similar to the one employed in this study.

This study has the potential for immediate translational impact. Since both antitumor antibodies and αCTLA-4 antagonists are approved and routinely used in the clinic, they could easily be combined with clinical-stage localized α4-1BB agonists. Indeed, our data demonstrated that even non-collagen anchored α4-1BB agonists synergize fairly well with antitumor antibodies in combination with αCTLA-4 therapy, identifying a potential triple combination therapy whose individual components are all already in clinical use. It would be of interest to further explore systemic versus local delivery of the αCTLA-4 component of this therapy, as previous studies have demonstrated that local delivery can both increase TdLN exposure while also reducing systemic toxicities[66]. Additionally, a collagen anchoring cytokine fusion utilizing human LAIR2 (one of two human variants of LAIR) as the collagen-binding domain (CLN-617) based off of prior work in our own lab is entering human trials later this year, demonstrating that LAIR is a collagen-binding domain with clinical relevance[27,28,93].

In conclusion, we found that effective TA99 + tumor localized α4-1BB-LAIR therapy requires a wave of de novo CD8+ T cell priming to achieve maximum efficacy. In this study, we generated this enhanced priming wave through whole CD4+ T cell compartment depletion with an αCD4 depleting antibody, Treg-specific ablation using Foxp3-DTR mice, or treatment with αCTLA-4, a modality known to increase priming. These combinations resulted in high levels of primary tumor efficacy, with ~80–100% complete response rates. However, only in the latter two strategies, which preserved CD4+ effector T cells, did mice also develop robust long-term immunological memory, with 100% of cured mice rejecting secondary tumor rechallenge. Our data demonstrate that at baseline, proper CD8+ T cell priming is constrained by Tregs present in the TdLN. All three priming enhancing strategies are directed towards Tregs, either depleting them completely (αCD4 and DT), or blocking their immunosuppressive pathways (αCTLA-4). This provides strong rationale for development of Treg-directed therapies that modulate Treg function in the TdLN which, in combination with proper immune agonists, can drive high levels of efficacy even in immunologically cold tumors.

## Methods
### Study design
All animal work in this study was conducted under the approval of the Massachusetts Institute of Technology Committee on Animal Care in accordance with federal, state, and local guidelines. The purpose of this study was to (i) evaluate the efficacy and safety of collagen anchoring α4-1BB-LAIR and subsequently to (ii) understand the

mechanism driving synergy between TA99 + α4-1BB-LAIR and αCD4 and finally to (iii) identity more clinically relevant therapies that synergize with TA99 + α4-1BB-LAIR. We used the syngeneic murine melanoma line B16F10 for all studies. Mice were randomized before beginning treatment to ensure equal tumor size in all groups and were monitored for tumor size and weight loss until euthanasia or until complete tumor regression. Investigators were not blinded during the studies. In all studies, there were at least 5 mice per experimental group, except for the bulk RNA-sequencing experiment which had 3-4 mice per group. No data/experiments were excluded unless there were technical issues with the experiment, and outliers were not excluded. Many experiments were repeated twice, and number of mice per group, number of experimental repeats, and statistical methods are noted in figure legends.

## Mice

C57Bl/6 (C57Bl/6NTac, B6-F) mice were purchased from Taconic. C57Bl/6 albino (B6(Cg)-Tyr$^{c-2J}$/J, #000058) mice were purchased from The Jackson Laboratory. C67Bl/6 Foxp3-DTR(B6.129(Cg)-Foxp3$^{tm3(DTR/GFP)Ayr}$/J, #016958) mice were a gift from the Spranger lab (MIT). C67Bl/6 OT-I mice (C57BL/6-Tg(TcraTcrb)1100Mjb/J, #003831) were a gift from the Irvine lab (MIT). B6 Foxp3-DTR mice and B6 OT-I mice were bred in house and genotyped using Transnetyx. All animal studies were conducted using female mice consistent with prior studies[27,28,67]. All mice were aged six to twelve weeks before start of study. For B6 Foxp3-DTR studies, wildtype control animals were purchased from Taconic. All mice were housed in a specific-pathogen free facility, fed normal chow and water ad libitum under standard animal facility conditions (12 h light/dark cycle, temperature of 22 °C, relative humidity of 40–70%) and were euthanized using CO2 asphyxiation.

## Cells

B16F10 cells were purchased from ATCC (CRL-6475). MC38 cells were a gift from J. Schlom, National Cancer Institute, Bethesda, MD. Apigmented B16F10 cells used for imaging were generated by genetic deletion of Tyrosinase-related-protein-2 (TRP2), referred to as B16F10-Trp2 KO cells[94]. Tumor cells were cultured in Dulbecco's Modified Eagle Medium (DMEM, ATCC) supplemented with 10% Fetal Bovine Serum (FBS, Gibco). FreeStyle 293-F cells and Expi293 cells were purchased from Invitrogen (R79007 and A14527, respectively) and cultured in FreeStyle expression medium (Gibco) and Expi293 expression medium (Gibco), respectively. CHO DG44 cells were a gift from David Hacker, EPFL, Lausanne, Switzerland. CHO DG44 cells were cultured in ProCHO5 (Lonza) supplemented with 4 mM L-glutamine, 0.1 mM hypoxanthine, and 16 μM thymidine. Tumor cells were maintained at 37 °C and 5% CO2 and FreeStyle 293-F cells, Expi293 cells, and CHO DG44 cells were maintained at 37 °C and 8% CO2. All cells tested negative for mycoplasma contamination.

## Tumor inoculation and treatment

Female mice were aged six to twelve weeks before tumor inoculations. $1 \times 10^6$ B16F10, B16F10-Trp2KO, or MC38 cells were suspended in 50uL sterile PBS (Corning) and injected subcutaneously on the right flank.

Mice were randomized before beginning treatment to ensure equal tumor size in all groups. TA99 was administered intraperitoneally (i.p) at a dose of 200 μg in 200 μL sterile PBS (Corning). α4-1BB or α4-1BB-LAIR was administered intratumorally (i.t.) in 20 μL of sterile PBS (Corning) at a dose of 30 μg or 36.1 μg (molar equivalents), respectively. αCD4 (Clone GK1.5, Bioxcell) was administered i.p. at a dose of 400 μg in 100 μL sterile PBS (Corning). αCTLA-4 (Clone 9D9, mIgG2c isotype) was administered i.p. at a dose of 200 μg in 100 μL of sterile PBS (Corning). Diphtheria Toxin (DT, Sigma Aldrich) was administered i.p. at a dose of 1 μg in 100 μL sterile PBS (Corning) or i.t. at a dose of 75 ng or 125 ng in 20 μL sterile PBS (Corning). Stock solutions of FTY720 (Sigma Aldrich) were resuspended at 10 mg/mL in DMSO and diluted to a dose of 30 μg in sterile PBS (Corning) to a final volume of 150 μL and administered i.p. For the MC38 study, 2.5F-Fc was administered i.p. at a dose of 400 μg in 200 μL sterile PBS (Corning) instead of TA99.

TA99 or 2.5F-Fc were dosed on days 5, 12, 19, and 26 and α4-1BB and α4-1BB-LAIR were administered on days 6, 13, 20, and 27. αCD4 was administered starting on day 4 and continued every three days until day 37 (Fig. 1a). For some studies, therapy initiation was delayed by 2 days to allow for larger tumors at time of analysis (flow cytometry, chemokine/cytokine analysis, and bulk-RNA-sequencing), sufficiently sized tumors for intratumoral DT administration (DT survival studies), or to avoid interfering with the endogenous T cell response (FTY720 studies, except Fig. S7C which followed Fig. 1a dosing scheme). DT was administered every other day starting on day 6 and continued until day 36. FTY720 was administered starting concurrently with αCD4 and continued every other day until one week after final α4-1BB-LAIR dose. "Delayed" FTY720 was administered starting concurrently with α4-1BB-LAIR and continued every other day until one week after the final α4-1BB-LAIR dose. αCTLA-4 was given on days 6, 9, 13, 16, 20, 23, and 27. For GM-CSF neutralization studies, αGM-CSF (Clone MP1-22E9, Bioxcell) was administered i.p. at a dose of 250 μg in 100 μL sterile PBS (Corning) starting on day 4 and continuing every other day until day 37.

During all tumor studies, mice were monitored continuously for tumor growth and weight change. Tumor growth was assessed by direct measurement with calipers and mice were euthanized when their tumor area (length × width) reached 100 mm$^2$ or mice lost more than 20% of their body weight, consistent with the protocol approved by the Massachusetts Institute of Technology Committee on Animal Care in accordance with federal, state, and local guidelines. Mice that were cured of their primary tumor but later euthanized due to overgrooming-related dermatitis were still classified as complete responders and included in analysis.

For rechallenge studies, mice that rejected their primary tumors were inoculated with $1 \times 10^5$ B16F10 tumor cells on the left, or contralateral, flank 100–110 days after primary tumor inoculation and monitored for tumor outgrowth. Age matched naïve mice were used as controls in these studies.

## Cloning and protein production

The heavy chain and light chain variable regions of α4-1BB antibody (clone LOB12.3) were synthesized as gBlock gene fragments (Integrated DNA technologies) and cloned into the gWiz expression vector (Genlantis) using In-fusion cloning (Takara Bio). Antibodies were expressed as chimeras with a murine kappa light chain constant region and a murine IgG1 heavy chain constant region. Antibodies were encoded in a single expression cassette with a T2A peptide inserted between the light chain and heavy chain. αFITC (clone 4420) were constructed in the same fashion, but a murine IgG2c isotype with LALA-PG silencing mutations was used for the heavy chain constant region[95]. For LAIR fusions, the murine LAIR1 gene was synthesized as a gBlock gene fragment (Integrated DNA technologies) and cloned as a fusion to the C-terminus of the heavy chain constant region separated by a flexible $(G_4S)_3$ linker. Plasmids were transformed into Stellar competent cells for amplification and isolated with Nucleobond Xtra endotoxin-free kits (Macherey-Nagel).

α4-1BB, aα4-1BB-LAIR, αFITC, and αFITC-LAIR were produced using the Expi293 expression system (Gibco) following manufacturer's instructions. Briefly, 1 mg/L of DNA and 3.2 mg/L of ExpiFectamine 293 were individually diluted into OptiMEM media (Gibco) and then combined dropwise. This mixture was then added dropwise to Expi293F suspension cells and 18–24 h later ExpiFectamine 293 Transfection enhancers 1 and 2 (Gibco) were added to the culture. 7 days after transfection, supernatants were harvested and antibodies were purified using Protein G sepharose 4 Fast Flow resin (Cytiva).

2.5F-Fc was produced using the FreeStyle 293-F expression system (Gibco). Briefly, 1 mg/L of DNA and 2 mg/L of polyethylenimine (Polysciences) were individually diluted into 20 mL/L OptiPro media (Gibco) and then combined dropwise. This mixture was then added dropwise to 293-F suspension cells (at a concentration of 1 M/mL) and 7 days after transfection, supernatants were harvested and antibodies were purified using rProtein A sepharose Fast Flow resin (Cytiva).

TA99 was produced using a FreeStyle 293-F stable production line generated in-house. Cells were expanded and then seeded at a density of 1 M/mL and supernatant was harvested 7 days later. 9D9 was produced using a CHO DG44 stable production line gifted to us by David Hacker, EPFL, Lausanne, Switzerland. Cells were expanded and then seeded at a density of 0.5 M/mL and supernatant was harvested 7 days later. Both TA99 and 9D9 were purified using rProtein A Sepharose Fast Flow resin (Cytiva).

Following purification, proteins were buffer exchanged into PBS (Corning) using Amicon Spin Filters (Sigma Aldrich), 0.22 µm sterile filtered (Pall), and confirmed for minimal endotoxin (<0.1 EU/dose) using the Endosafe LAL Cartridge Technology (Charles River). Molecular weight was confirmed with SDS-PAGE. Proteins run alongside a Novex Sharp Pre-Stained Protein Standard (Invitrogen) on a NuPAGE 4 to 12% Bis-Tris gel (Invitrogen) with 2-(N-morpholino) ethanesulfonic acid (MES) running buffer (VWR) and stained for visualization with SimplyBlue Safe Stain (Life Technologies). Proteins were confirmed to be free of aggregates by size exclusion chromatography using a Superdex 200 Increase 10/300 GL column on an Äkta Explorer FPLC system (Cytiva). All proteins were flash frozen in liquid nitrogen and stored at -80 °C.

## Collagen I ELISA
96 well plates precoated with rat collagen I (Gibco) were blocked overnight with PBSTA (PBS (Corning) + 2% w/v BSA (Sigma Aldrich) + 0.05% v/v Tween-20 (Millipore Sigma)) at 4 °C. After washing with 3 times PBST (PBS (Corning) + 0.05% v/v Tween-20 (Millipore Sigma)) and 3 times with PBS (Corning), α4-1BB and α4-1BB-LAIR were incubated in PBSTA overnight at 4 °C while shaking. Wells were washed 3 times with PBST and 3 times with PBS and then incubated with goat αmIgG1-Horseradish peroxidase (HRP) (1:2000, Abcam) in PBSTA for 1 h at RT while shaking. Wells were again washed 3 times with PBST and 3 times with PBS and then 1-Step Ultra TMB-ELISA Substrate Solution (Thermo Fisher) was added for 5–15 min, followed by 1 M sulfuric acid to quench the reaction. Absorbance at 450 nm (using absorbance at 570 nm as a reference) was measured on an Infinite M200 microplate reader (Tecan). Binding curves were generated with GraphPad Prism software V9. $K_D$ values were calculated using a nonlinear regression fit for one site total binding with no non-specificity and curves were normalized to the $B_{max}$ values.

## Surface 4-1BB binding assay
The gene for murine 4-1BB (OriGene) was cloned into the pIRES2 expression vector, which encodes for GFP downstream of the inserted 4-1BB gene using an IRES site, using In-Fusion cloning (Takara Bio). Freestyle 293-F cells were transiently transfected by mixing 1 mg/mL of plasmid DNA and 2 mg/mL of polyethylenimine (Polysciences) in OptiPRO Serum Free Medium (Gibco) and, after incubating, adding dropwise to the cells. 3-5 days after transfection, cells were harvested and pelleted in V-bottom 96 well plates. Cells were titrated with α4-1BB or α4-1BB-LAIR and incubated for 3 h shaking at 4 °C. Cells were washed with PBSA (PBS (Corning) + 0.1% BSA (Sigma Aldrich)) and incubated with αmIgG1-APC (diluted 1:250, clone M1-14D12, Invitrogen) for 30 min shaking at 4 °C. Data were collected on a BD LSR II cytometer (BD Biosciences). Binding curves were generated with GraphPad Prism software V9. $K_D$ values were calculated using a nonlinear regression fit for one site total binding with no non-specificity and curves were normalized to the $B_{max}$ values.

## OT-I splenocyte activation assay
Spleens were excised from OT-I mice and mechanically dissociated through a 70 µm filter and then red blood cells were removed using ACK lysis buffer (Gibco). Splenocytes were pulsed with 1 nM of G4 OVA peptide variant and indicated concentrations of either a4-1BB or a4-1BB-LAIR[96]. 3 ×10⁵ splenocytes were plated per well in 200 µL of media. Cells were cultured in Roswell Park Memorial Institute Medium (RPMI, ATCC) supplemented with 10% FBS (Gibco), 2mM L-glutamine (Gibco), 1X Non-essential amino acids (MEM-NEAA, Gibco), 1X Penicillin/Streptomycin (Gibco), 1X Sodium Pyruvate (Gibco), and 0.055 mM betamercaptoethanol (Gibco) at 37 °C and 5% CO2 in either non-TC treated flat bottom plates (Falcon) or collagen I coated flat bottom plates (Gibco). After 72 h supernatant was harvested and IFNγ levels were measured using the ELISA MAX Deluxe Set Mouse IFNγ kit (Biolegend). Binding curves were generated with GraphPad Prism software V9. EC50 values were calculated using a nonlinear regression fit for a three parameter agonist response curve.

## IVIS
Proteins were labeled with Alexa Fluor 647 NHS Ester (Life Technologies) and a Zeba desalting column (Thermo Scientific) was used to remove excess dye. The total molar amount of dye injected per sample was normalized between groups before injection. 20 µg of αFITC mIgG2c LALA-PG and a molar equivalent of αFITC-LAIR mIgG2c LALA-PG were used for in vivo retention studies. B6 albino mice were inoculated with 10⁶ B16F10-Trp2 KO cells and labeled proteins were injected i.t. on day 7. Fluorescence at the site of the tumor was measured longitudinally using the IVIS Spectrum Imaging System (Perkin Elmer). One week prior to study initiation, mice were switched to an alfalfa-free casein chow (Test Diet) to reduce background fluorescence. Total radiant efficiency was calculated after subtracting background fluorescence and normalizing to the maximum value for each protein using Living Image software (Caliper Life Sciences).

## Tumor cytokine/chemokine analysis
Tumors were excised, weighed, mechanically dissociated, and incubated in tissue protein extraction reagent (T-PER, Thermo Fisher Scientific) with 1% Halt protease and phosphatase inhibitors (Thermo Fisher Scientific) for 30 min at 4 °C while rotating. The lysates were then centrifuged and supernatants filtered through a Costar 0.22 µm SpinX filter (Corning) to remove any remaining debris. Lysates were flash frozen and stored at -20 °C until time of analysis. Lysates were analyzed with the 13-plex mouse cytokine release syndrome LEGENDplex panel and the Mouse/Rat Total/Active TGF-β1 LEGENDplex kit (Biolegend). Data were collected on a BD LSR II cytometer (BD Biosciences).

## Flow cytometry
Tumors were excised, weighed, and mechanically dissociated before being enzymatically digested using a gentleMACS Octo Dissociator with Heaters (Miltenyi Biotec) in gentleMACS C tubes (Miltenyi Biotec) and enzymes from the Mouse Tumor Dissociation Kit (Miltenyi Biotec). Tumors were digested using the 37C_m_TDK_1 program for soft tumors. Following digestion, tumors were filtered through a 40 µm filter and transferred to a V-bottom 96 well plate for staining. TdLN and spleens were excised, weighed, and mechanically dissociated through a 70 µm filter. Spleen samples were resuspended with 5 mL of ACK Lysis buffer (Gibco) to lyse red blood cells before being re-filtered through a 70 µm filter. TdLN and spleen samples were then transferred to a V-bottom 96 well plate for staining. Blood samples were collected via cardiac puncture into K3 EDTA-coated tubes (MiniCollect). 200 µL of blood was mixed with 1 mL of ACK lysis buffer (Gibco) to lyse red blood cells before being transferred to a V-bottom 96 well plate for staining. Precision Counting Beads (Biolegend) were added to each well to account for sample loss during processing and obtain accurate

counts. Cells were washed once with PBS and then resuspended in Zombie UV Fixable Viability Dye (Biolegend) to stain dead cells for 30 min at RT in the dark. Cells were then washed with FACS buffer (PBS (Corning) + 0.1% BSA (Sigma Aldrich) + 2 mM EDTA (Gibco)) and blocked with αCD16/CD32 antibody (Clone 93, eBioscience) for 20 min on ice in the dark and then stained for extracellular markers for 30 min on ice in the dark. Samples not requiring intracellular staining were washed with FACS buffer and fixed with BD Cytofix (BD Biosciences) for 30 min at RT in the dark. Cells were then washed and resuspended in FACS buffer. For samples requiring intracellular staining, cells were washed after extracellular staining, fixed and permeabilized with the Foxp3/Transcription Factor Staining Buffer Set (eBiosciences), and stained for 30 min at RT in the dark, before being washed and resuspended in FACS buffer. Samples were analyzed with a BD FACS Symphony A3 (BD Biosciences) and data was processed and analyzed with FlowJo V10. See Supplementary Fig. 10 for example gates.

Tumor and TdLN samples in Fig. 2 and S4 were stained in 100 μL with αCD45-BUV395 (30-F11, BD Biosciences #564279, 1:100 dilution), αCD4-BUV563 (RM4-4, BD Biosciences #741218, 1:100 dilution), αCD8α-BUV737 (53-6.7 BD Biosciences #612759, 1:100 dilution), αCD62L-BUV805 (MEL-14, BD Biosciences #741924, 1:100 dilution), αCD44-BV421 (1M7, Biolegend #103040, 1:100 dilution), αKi67-BV605 (16A8, Biolegend #652413, 1:100 dilution), αCD3-BV711 (17A2, Biolegend #100241, 1:100 dilution), αTIM-3-BV785 (RMT3-23, Biolegend #119725, 1:50 dilution), αTCF1/TCF7-AF488 (C63D9, Cell Signaling Technology #6444, 1:400 dilution), αPD-1-PerCp/Cy5.5 (29 F.1A12, Biolegend #135208, 1:50 dilution), αFoxp3-PE (FJK-16s, Invitrogen #12-5773-82, 1:200 dilution), αCD25-PE-Cy5 (PC61, Biolegend #102010, 1:100 dilution), αNK1.1-PE-Cy7 (PK-1366, Biolegend #108714, 1:100 dilution), α4-1BB-APC (17B5, Biolegend #106110, 1:50 dilution), αCD107a-APC-Cy7 (1D4B, Biolegend #121616, 1:100 dilution).

Tumor, TdLN, and spleen samples in Fig. 5a and S6 were stained in 100 μL with αCD45-BUV395 (30F-11, BD Bioscience #564279, 1:100 dilution), αCD8α-BUV737 (53-6.7, BD Biosciences #612759, 1:100 dilution), αCD3-BV785 (17A2, Biolegend #100232, 1:100 dilution), αNK1.1-PE-Cy7 (PK-136, Biolegend #108714, 1:100 dilution), αCD4-APC-Cy7 (GK1.5, Biolegend #100414, 1:100 dilution), and Foxp3+ cells were identified using the GFP reporter expressed under the Foxp3 locus in Foxp3-DTR mice.

Tumor, TdLN and blood samples in Fig. 5f, g and S7 were stained in 100 μL with αCD45-BUV395 (30-F11, BD Biosciences #564279, 1:100 dilution), αCD4-BUV563 (RM4-4, BD Biosciences #741218, 1:100 dilution), αCD44-BUV737 (1M7 BD Biosciences #612799, 1:100 dilution), αKi67-BV421 (16A8, Biolegend #652411, 1:100 dilution), αCD3-BV711 (17A2, Biolegend #100241, 1:100 dilution), αCD8α-FITC (53-6.7, Biolegend #100706, 1:100 dilution) αFoxp3-PE (FJK-16s, Invitrogen, 1:200 dilution), αCD25-PE-Cy5 (PC61, Biolegend #102010, 1:100 dilution), αNK1.1-PE-Cy7 (PK-136, Biolegend #108714, 1:100 dilution), αCD62L-APC (MEL-14, Biolegend #104412, 1:100 dilution), αCD107a-APC-Cy7 (1D4B, Biolegend #121616, 1:100 dilution).

## RNA extraction for sequencing

Tumor samples were processed as previously described. Samples were enriched for CD45+ cells using an EasySep Mouse TIL (CD45) Positive Selection kit (STEMCELL) and RNA was extracted with an RNeasy Plus Mini Kit (Qiagen). TdLN samples were processed as previously described. Samples were again enriched for CD45+ cells using an EasySep Mouse CD45 Positive Selection kit (STEMCELL) and RNA was extracted with an RNeasy Plus Mini Kit (Qiagen). RNA was stored at -80 °C until further processing.

## RNA-seq library preparation and sequencing

RNA-sequencing was performed by the BioMicro Center at MIT using a modified version of the SCRB-seq protocol[97]. Libraries were sequenced on a NextSeq 500 using a 75-cycle kit.

## RNA-seq alignment, quantification, and quality control

Data preprocessing and count matrix construction were performed using the Smart-seq2 Multi-Sample v2.2.0 Pipeline (RRID:SCR_018920) on Terra. For each cell in the batch, single-end FASTQ files were first processed with the Smart-seq2 Single Sample v5.1.1 Pipeline (RRID:SCR_021228). Reads were aligned to the GENCODE mouse (M21) reference genome using HISAT2 v2.1.0 with default parameters in addition to --k 10 options. Metrics were collected and duplicate reads marked using the Picard v.2.10.10 CollectMultipleMetrics and CollectRnaSeqMetrics, and MarkDuplicates functions with validation_stringency = silent. For transcriptome quantification, reads were aligned to the GENCODE transcriptome using HISAT2 v2.1.0 with --k 10 --no-mixed --no-softclip --no-discordant --rdg 99999999,99999999 --rfg 99999999,99999999 --no-spliced-alignment options. Gene expression was calculated using RSEM v1.3.0's rsem-calculate-expression --calc-pme --single-cell-prior. QC metrics, RSEM TPMs and RSEM estimated counts were exported to a single Loom file for each sample. All individual Loom files for the entire batch were aggregated into a single Loom file for downstream processing. The final output included the unfiltered Loom and the tagged, unfiltered individual BAM files.

## RNA-seq analysis

Samples with less than 10,000 genes detected were excluded from analysis. This led to exclusion of two tumor samples, one from the Tx + αCD4 group and one from the αCD4 group at the day 6 time point (Fig. S11). UMAP embedding of TdLN samples was generated from the top 5 principal components and top 3000 variable features. DEseq2 was used to conduct differential expression testing and apeglm was used for effect size estimation[98,99]. Pathways enrichment analysis for statistically significant upregulated genes was performed using enrichR to query the databases indicated in the text[57–59]. A score for the derived response gene signature was calculated for each experimental cohort using Seurat (AddModuleScore)[100]. Differential expression testing was performed as described above comparing all tumor sample cohorts to the D3 PBS, D6 PBS, D3 Tx + αCD4, and D6 Tx + αCD4 cohorts. All statistically significant hits (p-adj ≤5 with absolute value log2 fold-change ≥ 2 were included for further analysis. Gene clusters were defined using k-means clustering and the complexHeatmap package was used to generate expression heatmaps for these genes[101]. Relative expression profiles of these gene clusters were generated by summarizing the percent expression using Seurat (PercentageFeatureSet) per sample and dividing by the highest average percent per condition[100]. Gene sets were obtained from MSigDB and enrichment of genes from each cluster in these gene sets was calculated using the enrichGO function in the clusterProfiler package[102]. Gene Set Enrichment Analysis (GSEA) was performed using software from the Broad Institute (https://www.broadinstitute.org/gsea/index.jsp)[103,104]. Enrichment scores were calculated by comparing each treatment cohort (PBS, Tx, Tx + αCD4, αCD4) to the other three with 20,000 permutations. Gene signatures for SLEC and MPEC were obtained from GEO (GSE8678) or from gene signature supplement tables for the Tex signature, CD8-G, CD8-B, and CD8_1 through CD8_6[105–107].

## Statistical methods

Statistics were computed in GraphPad Prism v9 as indicated in figure captions. Survival studies were compared using the log-rank (Mantel-Cox) test. Flow data, tumor supernatant cytokine/chemokine data, and weight loss data were compared using one- or two-way ANOVA with Tukey's multiple comparison correction. Differential expression analysis using DESeq2 models counts for each gene using a negative binomial model and tests for significance using Wald tests[99]. Pathway enrichment analysis was calculated by Fisher's exact test. P values are corrected for multiple hypothesis testing using the Benjamini-

Hochberg procedure for all RNA-sequencing analysis. Sample size and *P*-value cutoffs are indicated in figure captions.

## Reporting summary
Further information on research design is available in the Nature Portfolio Reporting Summary linked to this article.

## Data availability
All sequencing data generated in this study can be found in the GEO database under accession GSE223087. All other data generated in this study are available in the paper or in the figshare repository associated with this study (https://doi.org/10.6084/m9.figshare.23805444). All materials available upon request. Source data are provided with this paper.

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

## Acknowledgements

J.R.P., B.M.L., L.S., and E.A.L., were supported by the NSF Graduate Research Fellowship Program. L.R.D was supported by the Ludwig Center at the Koch Institute. We thank the Koch Institute's Robert A. Swanson (1969) Biotechnology Center (National Cancer Institute Grant P30-CA14051) for technical support, specifically the Flow Cytometry Core Facility, Preclinical Imaging and Testing Facility, and the BioMicro Center. We thank the Spranger lab for the gift of the Foxp3-DTR breeding pair. We thank the Protein Production and Structure Core Facility at the École polytechnique fédérale de Lausanne for the development of the 9D9 stable line. Figures 1 and 7 were partially created with biorender.com. All materials are available upon request.

## Author contributions

J.R.P. and K.D.W conceived of study, designed experiments, and wrote the manuscript. J.R.P. conducted experiments and analyzed data. B.M.L. assisted with experiments and analyzed data. L.D., J.S., L.S., E.A.L., W.P. assisted with experiments. J.M.P. and B.D.B. assisted with analysis of RNA-Seq data.

## Competing interests

J.R.P. and K.D.W. are inventors on U.S. Provisional Patent application no. 62/738,981 regarding the aforementioned collagen-anchoring immunomodulatory molecules and methods thereof. Cullinan Oncology is the licensee of this patent. K.D.W. is a consultant/advisor for Cullinan Oncology. All other authors declare that they have no competing interests.
