## [Peer Review File · Nature Communications]

Tregs constrain CD8⁺ T cell priming required for curative intratumorally anchored anti-4-1BB immunotherapyREVIEWER COMMENTS

Reviewer #1 (Remarks to the Author):

The topic of the current study is local immunotherapy with collagen-anchored a4-1BB. Previously, the group published a study using collagen-anchored cytokines (Momin N. et al., *Science Translational Medicine* 2019) and the Hubbell lab published a study using systemic collagen-anchored a4-1BB (Ishihara J., *Science Translational Medicine* 2019).

Although the present study is grounded on collagen-anchored a4-1BB, the focus is more on the combination and the mechanism by which a4-1BB therapy can achieve high efficacy even in a poorly immunogenic tumor. The authors found that effective a4-1BB-therapy in the B16 model required triple combination with an anti-tumor antibody (TA99) and inhibition of Tregs (by aCTLA4) to allow for efficient CD8+ T cell priming in tdLNs. While the anchoring of a4-1BB was beneficial, the work gives the impression that this was ultimately only a secondary aspect.

The manuscript shows very convincing data. Everything is very detailed and easy to understand. The main point is how it is possible to achieve high efficacy with a4-1BB even in a weakly immunogenic tumor. A particular strength of the study is that also tumor rechallenge experiments were performed, which demonstrate memory and activity at distant sites, and are therefore especially important in the context of local immunotherapy. Overall a very nice work.

However, despite the compelling data, the added value of the study compared to the existing body of knowledge is not very convincingly presented in the current discussion section. The data need to be further contextualized with relevant related studies that are not currently cited (I am not associated with any of these studies). Apart from further highlighting the added value (partially also already in the introduction) some open questions (mentioned below) should be addressed in the discussion.

The points to be discussed are:

1. Synergy of a4-1BB with aCTLA4:

Such synergy was already shown (Kocak E. et al., *Cancer Research* 2006; not cited). Although this synergy in that study was not sufficient for B16, but only for MC38, still the synergistic interaction itself is not completely new and the combination of a4-1BB with aCTLA4 was therefore quite obvious. Interesting in the current study, however, is indeed that the triple combination with additionally TA99 leads to very high efficacy also in the B16 model.

2. The role of Tregs and priming in tdLNs:

This is cited and also nicely demonstrated in the current study (please also mention van Braeckel-Budimir N. et al., *Journal of Immunotherapy of Cancer* 2021, who showed the role of dtLNs in the context of a4-1BB/aOX40 plus aPD-L1). The confirmation of the concept in this clarity and current context is important, but also expected.

In this context, however, it is interesting to note that in the own previous study of the lab (Momin N. et al., *Science Translational Medicine* 2019), the combination of TA99 with locally administered collagen-anchored IL-2 showed high efficacy in the B16 model even WITHOUT inhibition of Tregs. The question therefore arises to what extent such very pronounced need for Treg inhibition in the present study might be a specific feature of a4-1BB therapy (resulting from activation and expansion of Tregs mediated by 4-1BB?; see also Kocak E. et al. 2006). Unfortunately, the authors do not address this aspect (they also do not discuss their earlier work) and it would be very interesting to know to what extent their present and earlier data provide clues here.

3. Relevance to clinical translation:

Importantly, there is already a clinical trial in which a4-1BB is administered locally. This trial (NCT03792724) needs to be cited. In addition, I also miss more concrete conclusions drawn from the results with regard to clinical implementation, especially the need for priming in tdLNs.

4. Furthermore, it would have been nice to see the direct comparison of especially a) the effect of aCTLA4 versus aPD-L1/aPD1:

Synergy of a4-1BB with aPD1 was shown in the B16 model (Chen S. et al., Cancer Immunology Research 2014; van Braeckel-Budimir N., Journal of Immunotherapy of Cancer 2021), furthermore, a4-1BB is frequently combined with aPD1/PD-L1 in clinical trials, including NCT03792724);

and possibly also b) systemic versus intratumoral aCTLA4:

The benefit of intratumoral administration of aCTLA4 is well known (Francis DM. et al., Science Translational Medicine 2020; meanwhile also in a clinical trial: van Pul KM. et al., Science Immunology 2022; intradermally administered after tumor excision; importantly before LN biopsy).

5. Inclusion of a few more details on collagen anchoring: Is the format used already the optimal version or what is ongoing in this area?; please include also a bit more discussion of the own work (Momin N. et al., Nature Communications 2022 and Science Translational Medicine 2019) and perhaps also some aspects of the own approach compared to systemic administration of collagen-anchored a4-1BB.

If word count is an issue, some more general paragraphs in the discussion could be shortened.

Some very small points:

Page 14, second-last paragraph: Wrong figure reference. "Fig. 7A" in "(Fig. 6G, Fig. 7A)." must be "Fig. S7A".

Fig. 6G => please include legend (same for Fig. S7F)

Fig. S7B => please change color code => lines hard to distinguish

Remove Fig. S8 and use as Fig. 7A instead (with changed color code for better distinction)

Reviewer #2 (Remarks to the Author):

- What are the noteworthy results?

In this study, the authors examine anti-tumor efficacy and underlying mechanism of agonistic rat anti-mouse IgG1 antibody to 4-1BB (LOB12.3) conjugated (or not) to the collagen binding domain of the receptor LAIR. They validate tumor retention of the a4-1BB-LAIR conjugate as compared to the native form and find that this conjugate is the better therapeutic in a specific mouse tumor setting.

- Will the work be of significance to the field and related fields? How does it compare to the established literature? If the work is not original, please provide relevant references.

Agonistic antibodies to 4-1BB are in clinical testing as immunotherapeutic drugs, but have been disappointing because of systemic toxicity. Toxicity in human cannot be properly predicted from mouse models, in part because of differential expression of 4-1BB (CD137) and because of differential distribution of Fc receptors. The authors show in the difficult to treat mouse melanoma tumor model B16F10, that anti-4-1BB-LAIR is more efficacious than anti-4-1BB in terms of tumor elimination, provided that a Treg response is prevented. They also show convincingly that in this model, Tregs constrain the CTL response and thereby the therapeutic effect of the antibodies. The study is novel in regards to the therapeutic effects of this newly generated reagent, but it is not novel in terms of the constraint of Tregs on generating CTL activity against this model tumor.

- Does the work support the conclusions and claims, or is additional evidence needed?

Are there any flaws in the data analysis, interpretation and conclusions? Do these prohibit publication or require revision?

This is a serious study that is conducted according to good scientific standards. It is correct in its design and honest in its interpretations. It is a bit deficient in the low complexity of T-cell response analysis.

Main comments:

- 1) It should be much more explicitly stated that the parent anti-4-1BB antibody is an agonist. I found out that it is sold as such by BioXcell. This is important, because not all antibodies are agonists and we want to know whether adding the collagen binding domain and increased tumor retention is the key reason why the anti-4-1BB-LAIR conjugate is more effective than the parent antibody.
- 2) The authors should add an in vitro assay, wherein in presence or absence of a collagen layer, the T-cell costimulatory activity of the parent antibody and the anti-4-1BB-LAIR conjugate are compared.
- 3) The key reason why the 4-1BB-LAIR conjugate is the better reagent might be because collagen binding makes it a (better) agonist. For this reason, I am currently not convinced that tumor retention rather than increased agonist activity is the reason that the anti-4-1BB-LAIR conjugate is the better therapeutic. This is particularly so because the effect is dependent on new T-cell priming.

The authors should test whether the 4-1BB-LAIR conjugate can act locally in the tumor, I suggest in a simpler model. They can use MC-38, which is well-defined immunogenic model that responds to anti-PD-1 therapy. If tumors are large, they respond to anti-PD-1 therapy, even when LN communication is blocked by FTY-720. I would like to see in such a setting, representing T-cell infiltrated cancers, where there is no Treg constraint on CTL priming, if there is a difference in anti-4-1BB-LAIR versus parental antibody treatment.

- 4) The authors use bulk RNAseq to look at the T cell response, which is on one hand commendable, but lacks quite a lot of detail on the CTL response. It would be much better if they look at the treatment effects of the antibody and control with flow cytometry in regards to tumor-antigen-specific stem-like versus terminally differentiated CD8 effector T cells. This can be done by flow cytometry with MHC tetramers and appropriate antibodies in multicolor flow cytometry in both the B16 and the MC-38 model. I suggest they do this in the anti-CTLA4 combination setting when using B16.

Other comments:

- 1) The authors should explain why CD4 T cell depletion is not a desired approach. CD4 T cells are very important for optimization of CTL effector and memory quality through CD4 T-cell help. This is the reason why the authors do not get CTL memory after CD4 T cell depletion but do get memory after selective Treg depletion or anti-CTLA4 treatment.

- Is the methodology sound? Does the work meet the expected standards in your field?
Is there enough detail provided in the methods for the work to be reproduced?

The methodology is sound and well described.

Reviewer #3 (Remarks to the Author):

In this manuscript, Palmeri and colleagues developed a new tumor-anchored anti-4-1BB antibody (a4-1BB-LAIR) to increase anti-4-1BB's on-target effects and reduce off-tumor toxicity. When a4-1BB-LAIR was used in combination with an anti-tumor antibody TA99, the anti-tumor effect was moderate in the B16 melanoma mouse model. However, the anti-tumor effect of the a4-1BB-LAIR+TA99 treatment was significantly enhanced when CD4 T cells were depleted. This enhancement of the anti-tumor effect with anti-CD4 antibody treatment was due to the depletion of the Treg populations. Finally, the combination of a4-1BB-LAIR+TA99 and anti-CTLA-4 antibody showed the equivalent anti-tumor effect compared to the anti-CD4 antibody, and generated lasting immunity against secondary tumor challenge.

The development of the tumor-anchored anti-4-1BB antibody is exciting, and the result from the combination of a4-1BB-LAIR+TA99+aCTLA-4 treatment showed the potential of this new strategy that could be applicable to human cancer. However, several concerns dampened the enthusiasm for the study, as listed below.

1. The structure of the current manuscript does not match the importance of its findings proportionally. The CD4 T cell depletion experiment, which is currently covered in Figures 2 to 5, should be a much smaller part of the manuscript. The aCTLA-4 combination treatment experiment is more exciting and important. Yet it was only presented briefly in Fig. 7. This study needs to be expanded significantly to the level of the CD4 depletion experiment, with detailed characterization of the immune cell profiles.
2. The authors concluded from the CD4 depletion experiment that CD8 T cell memory response requires CD4 T helper cells. This is well-known in the T cell memory field. The original works on this topic were published 20 years ago (Sun JC et al, Science, 2003, PMID: 12690202; Janssen EM et al, Nature, 2003, PMID: 12594515)
3. The authors performed bulk RNA-seq of all CD45+ cells from TDLNs and tumors. This is a very crude approach because CD45+ include all immune cells. Therefore the RNA-seq cannot reliably track gene expression changes in different immune cell subsets. Instead of bulk RNA-seq, single cell RNA-seq should be performed to accurately track immune cell dynamics. At the minimum, bulk RNA-seq should be performed with sorted CD8 T cells since they are the driver of the anti-tumor immune response in this model.
4. In Fig. 6A, low dose i.t. DT injection still dramatically reduced spleen Treg population by ~80%. This will induce systemic T cell activation that influences anti-tumor immunity indirectly.
5. In the anti-CTLA-4 combination therapy experiment shown in Fig. 7, anti-CTLA-4 antibody treatment alone without Tx should be a critical control to be included.
6. It has been shown that the anti-CTLA-4 therapy partially depleted intra-tumor Tregs (Simpson TR et al, JEM, 2013, PMID: 23897981.). This work is highly relevant to the current study and should be cited.

Reviewer #1 (Remarks to the Author):

The topic of the current study is local immunotherapy with collagen-anchored a4-1BB. Previously, the group published a study using collagen-anchored cytokines (Momin N. et al., Science Translational Medicine 2019) and the Hubbell lab published a study using systemic collagen-anchored a4-1BB (Ishihara J., Science Translational Medicine 2019).

Although the present study is grounded on collagen-anchored a4-1BB, the focus is more on the combination and the mechanism by which a4-1BB therapy can achieve high efficacy even in a poorly immunogenic tumor. The authors found that effective a4-1BB-therapy in the B16 model required triple combination with an anti-tumor antibody (TA99) and inhibition of Tregs (by aCTLA4) to allow for efficient CD8+ T cell priming in tdLNs. While the anchoring of a4-1BB was beneficial, the work gives the impression that this was ultimately only a secondary aspect.

The manuscript shows very convincing data. Everything is very detailed and easy to understand. The main point is how it is possible to achieve high efficacy with a4-1BB even in a weakly immunogenic tumor. A particular strength of the study is that also tumor rechallenge experiments were performed, which demonstrate memory and activity at distant sites, and are therefore especially important in the context of local immunotherapy. Overall a very nice work.

We thank the reviewer for their kind comments regarding the data and presentation of the manuscript, particularly their appreciation of the rechallenge studies!

However, despite the compelling data, the added value of the study compared to the existing body of knowledge is not very convincingly presented in the current discussion section. The data need to be further contextualized with relevant related studies that are not currently cited (I am not associated with any of these studies). Apart from further highlighting the added value (partially also already in the introduction) some open questions (mentioned below) should be addressed in the discussion.

The points to be discussed are:

1. Synergy of a4-1BB with aCTLA4:

Such synergy was already shown (Kocak E. et al., Cancer Research 2006; not cited). Although this synergy in that study was not sufficient for B16, but only for MC38, still the synergistic interaction itself is not completely new and the combination of a4-1BB with aCTLA4 was therefore quite obvious. Interesting in the current study, however, is indeed that the triple combination with additionally TA99 leads to very high efficacy also in the B16 model.

We thank the reviewer for pointing out this study and have included a sentence citing this prior literature.

2. The role of Tregs and priming in tdLNs:

This is cited and also nicely demonstrated in the current study (please also mention van Braeckel-Budimir N. et al., Journal of Immunotherapy of Cancer 2021, who showed the role of dtLNs in the context of a4-1BB/aOX40 plus aPD-L1). The confirmation of the concept in this clarity and current context is important, but also expected.

In this context, however, it is interesting to note that in the own previous study of the lab (Momin N. et al., Science Translational Medicine 2019), the combination of TA99 with locally administered collagen-anchored IL-2 showed high efficacy in the B16 model even WITHOUT inhibition of Tregs. The question therefore arises to what extent such very pronounced need for Treg inhibition in the present study might be a specific feature of a4-1BB therapy (resulting from activation and expansion of Tregs mediated by 4-1BB?; see also Kocak E. et al. 2006). Unfortunately, the authors do not address this aspect (they also do not discuss their earlier work) and it would be very interesting to know to what extent their present and earlier data provide clues here.

We thank the reviewer for pointing out this additional citation regarding the role of TdLNs in the context of anti-4-1BB therapy. However, we respectfully disagree that this study is a fair comparison to our own work. In the study suggested by the reviewer, the authors demonstrated that anti-OX40 + anti-4-1BB expanded stem-like T cells in the TdLN and then anti-PD-L1 further expanded and differentiated these cells towards a more terminally exhausted state to drive therapeutic efficacy. Although they nicely demonstrated that this mechanism involves the TdLN, it differs from our study on two key points - 1) the authors only observed this effect with the combination therapy anti-OX40 and anti-4-1BB, not anti-4-1BB on its own and 2) they showed that the effect of the anti-4-1BB agonist itself is in the TdLN, whereas our RNA-seq and

flow data suggests that our collagen anchored anti-4-1BB agonist's effects are restricted to the tumor, and it is the anti-CD4/DT/anti-CTLA-4 that has TdLN effects. Thus, although both studies involve anti-4-1BB agonists with some TdLN involvement they are fundamentally different mechanisms of action in our opinion.

Comparisons to our early work looking at TA99 + Lumican-MSA-IL2 are indeed interesting. Unfortunately, we do not have mechanistic data for this combination so it is difficult to make concrete statements about how this combination therapy works. It is quite possible this is a feature specific to anti-4-1BB therapy (or agonist antibody therapy broadly). However, due to the lack of mechanistic data on TA99 + Lumican-MSA-IL2 any comparisons between this manuscript and that prior work would be highly speculative and we are uncomfortable making such speculative statements in our discussion section.

3. Relevance to clinical translation:

Importantly, there is already a clinical trial in which a4-1BB is administered locally. This trial (NCT03792724) needs to be cited. In addition, I also miss more concrete conclusions drawn from the results with regard to clinical implementation, especially the need for priming in tdLNs.

We thank the reviewer for pointing out this critical clinical trial and have included it in the discussion section on intratumoral administration. To clarify, we interpret our data to suggest that the anti-4-1BB-LAIR agonist antibody is needed in the tumor, and the "priming agents" used in this study (namely anti-CD4 and anti-CTLA-4) act on the TdLN. These agents were given systemically, as currently done in the clinic, and such the need for priming in the TdLN should not impact the clinical implementation of our proposed therapy - put simply we do not view a need for intranodal or perinodal injections of any of the agents to achieve the lymph node priming effect. Additionally, as the reviewer has nicely pointed out with citations in the next comment, local delivery of anti-CTLA-4 is 1) already in the clinic and 2) achievable (albeit to a lesser extent) with intratumoral administration. A short paragraph on intratumoral administration is included in the discussion.

4. Furthermore, it would have been nice to see the direct comparison of especially a) the effect of aCTLA4 versus aPD-L1/aPD1:

Synergy of a4-1BB with aPD1 was shown in the B16 model (Chen S. et al., Cancer Immunology Research 2014; van Braeckel-Budimir N., Journal of Immunotherapy of Cancer 2021), furthermore, a4-1BB is frequently combined with aPD1/PD-L1 in clinical trials, including NCT03792724);

and possibly also b) systemic versus intratumoral aCTLA4:

The benefit of intratumoral administration of aCTLA4 is well known (Francis DM. et al., Science Translational Medicine 2020; meanwhile also in a clinical trial: van Pul KM. et al., Science Immunology 2022; intradermally administered after tumor excision; importantly before LN biopsy).

We thank the reviewer for these suggestions. Although anti-CTLA-4 and anti-PD1/anti-PD-L1 are both ICB therapies, recent literature suggests that they have fairly different mechanisms of

action. We chose anti-CTLA-4 because of its ability to enhance *de novo* priming of immune responses. Anti-PD-1, on the other hand, seems to primarily work by allowing for the expansion and differentiation of stem-like CD8+ T cells that have already been primed. In fact, some evidence would suggest that giving anti-PD-1 before/during T cell priming (such as dosing anti-PD-1 three days prior to vaccination, as demonstrated in Verma et al., Nature Immunology 2018) actually *impairs* the overall anti-tumor immune response with accompanying increases in T cell apoptosis. Thus, given what we know about the mechanisms of anti-CTLA-4, anti-PD-1, and our understanding of anti-CD4's role in the context of this manuscript, we do not see strong rationale for combining anti-PD-1 with TA99 + anti-4-1BB-LAIR. Of course, as the reviewer has pointed out this synergy has already been shown and anti-4-1BB (along with most immunotherapy agents) has been combined with anti-PD-1/anti-PD-L1 in the clinic, but we do not believe that this combination fits mechanistically with the present manuscript.

The comparison of intratumoral vs. systemic anti-CTLA-4 is certainly interesting. We have added language to the discussion section suggesting exploring systemic vs. intratumoral as follow up work. Ultimately, we have decided to focus all of our mechanistic understandings of this therapeutic strategy on the Tx + anti-CD4 combination as this is where we first observed the dramatic synergy of this combination therapy. The Tx + anti-CTLA-4 combination is certainly more exciting from a translational perspective, but in our opinion any mechanistic studies (and further optimization) of this combination constitute follow up work outside the scope of this manuscript. Further optimization of the triple combination of antitumor antibody + anti-4-1BB-LAIR + anti-CTLA-4 would be necessary for further clinical translation and exploring intratumoral administration would be worthwhile as it would 1) limit toxicities driven by blockade in distant sites and 2) result in enhanced accumulation in TdLNs, according to the references provided by the reviewer.

5. Inclusion of a few more details on collagen anchoring: Is the format used already the optimal version or what is ongoing in this area?; please include also a bit more discussion of the own work (Momin N. et al., Nature Communications 2022 and Science Translational Medicine 2019) and perhaps also some aspects of the own approach compared to systemic administration of collagen-anchored a4-1BB.

We have not explored the use of higher affinity collagen anchoring domains (such as the LAIR-LAIR dimer domains used in Nature Communications 2022) so any commentary would be speculative. Certainly for clinical translation it would be important to optimize dose, dosing scheme, and collagen binding affinity but that is outside the scope of this academic study. Our prior work on collagen anchored cytokines is currently being developed by Cullinan Oncology and their current clinical lead (publicly presented at SITC in both 2021 and 2022; Mehta et al., JITC 2021 and Mehta et al., JITC 2022) contains a single human LAIR binding domain, so we can confidently say that the collagen anchoring format used in this study is certainly clinically relevant. We have added language to the discussion section regarding LAIR in the clinic. We have not tested systemic delivery of our constructs but, given the collagen binding profile of LAIR, it is likely to largely stick at the site of injection and/or in the vasculature after injection.

The study pointed out by the reviewer (Ishihara J., Science Translational Medicine 2019) does not cover systemically delivered collagen-anchored anti-4-1BB agonists, but rather systemically delivered collagen-anchored checkpoint blockade therapies. To our knowledge, this is the first report of collagen-anchored anti-4-1BB agonists in the literature. Given the growing feasibility of intratumoral injections, we are generally in favor of employing strategies to retain payloads directly injected at the site of the tumor vs. systemic delivery of “tumor targeting” payloads.

If word count is an issue, some more general paragraphs in the discussion could be shortened.

Some very small points:

Page 14, second-last paragraph: Wrong figure reference. “Fig. 7A” in “(Fig. 6G, Fig. 7A).” must be “Fig. S7A”.

Fig. 6G => please include legend (same for Fig. S7F)

Fig. S7B => please change color code => lines hard to distinguish

Remove Fig. S8 and use as Fig. 7A instead (with changed color code for better distinction)

We thank the reviewer for their thorough reading (and catching our typos!) and have incorporated their feedback into our figure edits. Note that the legends for 6G and 7F are already present in the figure for analogous flow plots and were omitted to prevent overcrowding of the figure as they seemed redundant.

Reviewer #2 (Remarks to the Author):

- What are the noteworthy results?

In this study, the authors examine anti-tumor efficacy and underlying mechanism of agonistic rat anti-mouse IgG1 antibody to 4-1BB (LOB12.3) conjugated (or not) to the collagen binding domain of the receptor LAIR. They validate tumor retention of the a4-1BB- LAIR conjugate as compared to the native form and find that this conjugate is the better therapeutic in a specific mouse tumor setting.

- Will the work be of significance to the field and related fields? How does it compare to the established literature? If the work is not original, please provide relevant references.

Agonistic antibodies to 4-1BB are in clinical testing as immunotherapeutic drugs, but have been disappointing because of systemic toxicity. Toxicity in human cannot be properly predicted from mouse models, in part because of differential expression of 4-1BB (CD137) and because of differential distribution of Fc receptors. The authors show in the difficult to treat mouse melanoma tumor model B16F10, that anti-4-1BB-LAIR is more efficacious than anti-4-1BB in terms of tumor elimination, provided that a Treg response is prevented. They also show convincingly that in this model, Tregs constrain the CTL response and thereby the therapeutic effect of the antibodies.

The study is novel in regards to the therapeutic effects of this newly generated reagent, but it is not novel in terms of the constraint of Tregs on generating CTL activity against this model tumor.

- Does the work support the conclusions and claims, or is additional evidence needed? Are there any flaws in the data analysis, interpretation and conclusions? Do these prohibit publication or require revision?

This is a serious study that is conducted according to good scientific standards. It is correct in its design and honest in its interpretations. It is a bit deficient in the low complexity of T- cell response analysis.

We thank the reviewer for their kind comments regarding the study design and interpretation of the results.

Main comments:

1) It should be much more explicitly stated that the parent anti-4-1BB antibody is an agonist. I found out that it is sold as such by BioXcell. This is important, because not all antibodies are agonists and we want to know whether adding the collagen binding domain and increased tumor retention is the key reason why the anti-4-1BB-LAIR conjugate is more effective than the parent antibody.

We have clarified the text to indicate that the parent anti-4-1BB clone used in this paper is an agonistic antibody. We have also added a second citation, in addition to the one already present, to support this point.

2) The authors should add an *in vitro* assay, wherein in presence or absence of a collagen layer, the T-cell costimulatory activity of the parent antibody and the anti-4-1BB-LAIR conjugate are compared.

We thank the reviewer for suggesting this informative assay and we have included it in the supplement. Notably, we found that in an *in vitro* OT-1 splenocyte activation assay (using interferon gamma secretion as a read out of activation) we found that anti-4-1BB and anti-4-1BB-LAIR had similar EC_{50} values in both the presence and absence of collagen, indicating that in this system there is no evidence that the presence of collagen can enhance the agonistic activity of anti-4-1BB-LAIR agonist antibodies.

We have

3) The key reason why the 4-1BB-LAIR conjugate is the better reagent might be because collagen binding makes it a (better) agonist. For this reason, I am currently not convinced that tumor retention rather than increased agonist activity is the reason that the anti-4-1BB-LAIR conjugate is the better therapeutic. This is particularly so because the effect is dependent on new T-cell priming.

The authors should test whether the 4-1BB-LAIR conjugate can act locally in the tumor, I suggest in a simpler model. They can use MC-38, which is well-defined immunogenic model that responds to anti-PD-1 therapy. If tumors are large, they respond to anti-PD-1 therapy, even when LN communication is blocked by FTY-720. I would like to see in such a setting, representing T-cell infiltrated cancers, where there is no Treg constraint on CTL priming, if there is a difference in anti-4-1BB-LAIR versus parental antibody treatment.

We have added additional survival data to the supplement demonstrating that the general paradigm of Tx alone or anti-CD4 alone provides some level of tumor control, but Tx + anti-CD4 having the highest level of efficacy also holds true in the MC38 model, although the results are not as dramatic (fewer overall survivors). In this combination, we used 2.5F-Fc as an antitumor antibody-like molecule in place of TA99, which we have previously published on (Kwan et al., J Exp Med 2017).

We have also included below monotherapy data comparing anti-4-1BB to anti-4-1BB-LAIR in the MC38 model as a monotherapy (mice were dosed with 30ug or 36.1ug (molar equivalent) of anti-4-1BB or anti-4-1BB-LAIR intratumorally on days 6, 12, and 18). Both treatments provided a statistically significant growth delay compared to PBS, although there is no statistically significant difference between the two groups (also comparing absolute number of survivors using Fisher's exact test does not result in a statistically significant difference between anti-4-1BB or anti-4-1BB-LAIR ($P = 1$)). Because this monotherapy data is on a slightly different dosing schedule than our manuscript we prefer to include it only here in the reviewer response.

4) The authors use bulk RNAseq to look at the T cell response, which is on one hand commendable, but lacks quite a lot of detail on the CTL response. It would be much better if they look at the treatment effects of the antibody and control with flow cytometry in regards to tumor-antigen-specific stem-like versus terminally differentiated CD8 effector T cells. This can be done by flow cytometry with MHC tetramers and appropriate antibodies in multicolor flow cytometry in both the B16 and the MC-38 model. I suggest they do this in the anti-CTLA4 combination setting when using B16.

We have included additional flow cytometry data looking at stem like vs. terminally differentiated CD8+ T cells (defined as TCF1+ Tim-3- and TCF1- Tim-3+, respectively, and pregated on PD-1+ CD8+ T cells) in the Tx + anti-CD4 combination in the B16F10 model. We do not observe

any major shifts in the populations among the various treatment groups when looking in the TdLN.

Although we agree that adding a tetramer stain would enhance the understanding of the overall CTL response, we believe it is not necessary to support the claims made in our paper. Additionally, we have decided to focus all of our mechanistic understandings of this therapeutic strategy on the Tx + anti-CD4 as this is where we first observed the dramatic synergy of this combination therapy. The Tx + anti-CTLA-4 combination is certainly more exciting from a translational perspective, but in our opinion any mechanistic studies of this combination constitute follow up work beyond the scope of this manuscript.

We have also performed additional analysis of our bulk RNA-seq data to further support our claims. Specifically, we analyzed our tumor sequencing using CIBERSORTx (Newman et al., *Nat Biotechnol* (2019)), a deconvolution algorithm aimed at estimating cell type abundance in bulk RNA-seq data (utilizing the mouse signature matrix published by Chen et al., *Sci Rep* (2017)), and using GSEA to assess enrichment of various published CD8 T cell signatures.

CIBERSORT analysis was partially consistent with our results, and we see high abundance of both total T cells and total CD8 T cells in the Tx + anti-CD4 group (And to a lesser extent in the anti-CD4 only group). Unfortunately, likely due to know drop out issues associated with *Cd4* transcripts, a large portion of T cells were misclassified as CD4⁺ T cells, which we know to be untrue as we consistently observe complete CD4⁺ T cell depletion with our therapy (notably, we also see a higher proportion of CD4 T cells in the Tx + anti-CD4 group). Because of this obvious error encountered using this analysis pipeline, we feel that this data should not be included in the manuscript but have below in this reviewer response. Interestingly, we also observed a decrease in relative fractions of both monocytes and M1 macrophages in the Tx + anti-CD4 group. We have included the results below and also attached the full output of the CIBERSORT analysis to this response.

We performed Gene Set Enrichment Analysis (GSEA) using software from the broad institute (<https://www.gsea-msigdb.org/gsea/index.jsp>), focusing on day 6 tumor samples, as this is when we saw the largest changes to the CD8⁺ T cell compartment in the tumor. The results of this analysis can be seen in extended data figure 6 and full details of the analysis and citations for gene-sets used in analysis can be found in the methods section. We found that while both PBS (and Tx only) were significantly de-enriched for CD8 T cell clusters associated with activation, memory, and response to immunotherapy, the Tx + anti-CD4 cohort was significantly and highly enriched for these CD8⁺ T cell gene-sets. Additionally, we observed that while Tx or anti-CD4 on their own drove enrichment of a CD8⁺ T cell exhaustion signature, the combination of Tx + anti-CD4 did not. Overall, this new analysis further supports our claim that only the Tx + anti-CD4 therapy drives a robust cytotoxic T cell program in the tumor.

Other comments:

1) The authors should explain why CD4 T cell depletion is not a desired approach. CD4 T cells are very important for optimization of CTL effector and memory quality through CD4 T-cell help. This is the reason why the authors do not get CTL memory after CD4 T cell depletion but do get memory after selective Treg depletion or anti-CTLA4 treatment.

We have added additional language to the discussion section detailing why whole CD4 T cell depletion is not desirable with respect to memory formation, in addition to what is already included in the discussion, and included several references on the topic of the role of CD4 T cell help in memory formation suggested by reviewer 3.

- Is the methodology sound? Does the work meet the expected standards in your field? Is there enough detail provided in the methods for the work to be reproduced?

The methodology is sound and well described.

We thank the reviewer for their kind comments regarding the methodology of the study.

Reviewer #3 (Remarks to the Author):

In this manuscript, Palmeri and colleagues developed a new tumor-anchored anti-4-1BB antibody (a4-1BB-LAIR) to increase anti-4-1BB's on-target effects and reduce off-tumor toxicity. When a4-1BB-LAIR was used in combination with an anti-tumor antibody TA99, the anti-tumor effect was moderate in the B16 melanoma mouse model. However, the anti-tumor effect of the a4-1BB-LAIR+TA99 treatment was significantly enhanced when CD4 T cells were depleted. This enhancement of the anti-tumor effect with anti-CD4 antibody treatment was due to the depletion of the Treg populations. Finally, the combination of a4-1BB-LAIR+TA99 and anti-CTLA-4 antibody showed the equivalent anti-tumor effect compared to the anti-CD4 antibody, and generated lasting immunity against secondary tumor challenge.

The development of the tumor-anchored anti-4-1BB antibody is exciting, and the result from the combination of a4-1BB-LAIR+TA99+aCTLA-4 treatment showed the potential of this new strategy that could be applicable to human cancer. However, several concerns dampened the enthusiasm for the study, as listed below.

We thank the reviewer for their positive comments regarding the translatability of the TA99 + anti-4-1BB-LAIR + anti-CTLA-4 combination.

1. The structure of the current manuscript does not match the importance of its findings proportionally. The CD4 T cell depletion experiment, which is currently covered in Figures 2 to 5, should be a much smaller part of the manuscript. The aCTLA-4 combination treatment experiment is more exciting and important. Yet it was only presented briefly in Fig. 7. This study needs to be expanded significantly to the level of the CD4 depletion experiment, with detailed characterization of the immune cell profiles.

Although we agree that the results of TA99 + anti-4-1BB-LAIR + anti-CTLA-4 are the most exciting and important from a clinical translation perspective, we respectfully disagree with this comment advising us to restructure the manuscript. We have presented the story here as it was carried out - initial observation of Tx + anti-CD4 efficacy, mechanistic study of synergy, and demonstration that the uncovered mechanistic paradigm still holds with a more clinically relevant modality. It is our opinion that mechanistic studies of the anti-CTLA-4 combination represent future work and are outside the scope of the story as presented.

2. The authors concluded from the CD4 depletion experiment that CD8 T cell memory response requires CD4 T helper cells. This is well-known in the T cell memory field. The original works on this topic were published 20 years ago (Sun JC et al, Science, 2003, PMID: 12690202; Janssen EM et al, Nature, 2003, PMID: 12594515)

We thank the reviewer for providing these references. We did not intend to suggest that this was a novel observation (it felt intuitively obvious, actually), and we have updated the language to clarify this and cite this prior literature.

3. The authors performed bulk RNA-seq of all CD45+ cells from TDLNs and tumors. This is a very crude approach because CD45+ include all immune cells. Therefore the RNA-seq cannot reliably track gene expression changes in different immune cell subsets. Instead of bulk RNA-seq, single cell RNA-seq should be performed to accurately track immune cell dynamics. At the minimum, bulk RNA-seq should be performed with sorted CD8 T cells since they are the driver of the anti-tumor immune response in this model.

Although we agree that scRNA-seq or bulk RNA-seq on sorted CD8+ T cells would provide more granularity on the immune cell dynamics, we feel that it is outside the scope of the study and, furthermore, the claims made in the manuscript are supported by the current bulk RNA-seq data. We have conducted additional orthogonal analysis of the bulk RNA-seq data to further support our claims. Specifically, we analyzed our tumor sequencing using CIBERSORTx (Newman et al., *Nat Biotechnol* (2019) PMID: 31061481), a deconvolution algorithm aimed at estimating cell type abundance in bulk RNA-seq data (utilizing the mouse signature matrix published by Chen et al., *Sci Rep* (2017) PMID: 28084418), and using GSEA to assess enrichment of various published CD8 T cell signatures.

CIBERSORT analysis was partially consistent with our results, and we see high abundance of both total T cells and total CD8 T cells in the Tx + anti-CD4 group (And to a lesser extent in the anti-CD4 only group). Unfortunately, likely due to know drop out issues associated with *Cd4* transcripts, a large portion of T cells were misclassified as CD4+ T cells, which we know to be untrue as we consistently observe complete CD4+ T cell depletion with our therapy (notably, we also see a higher proportion of CD4 T cells in the Tx + anti-CD4 group). Because of this obvious error encountered using this analysis pipeline, we feel that this data should not be included in the manuscript but have below in this reviewer response. Interestingly, we also observed a decrease in relative fractions of both monocytes and M1 macrophages in the Tx + anti-CD4 group. We have included the results below and also attached the full output of the CIBERSORT analysis to this response.

We performed Gene Set Enrichment Analysis (GSEA) using software from the broad institute (<https://www.gsea-msigdb.org/gsea/index.jsp>), focusing on day 6 tumor samples, as this is when we saw the largest changes to the CD8⁺ T cell compartment in the tumor. The results of this analysis can be seen in extended data figure 6 and full details of the analysis and citations for gene-sets used in analysis can be found in the methods section. We found that while both PBS (and Tx only) were significantly de-enriched for CD8 T cell clusters associated with activation, memory, and response to immunotherapy, the Tx + anti-CD4 cohort was significantly and highly enriched for these CD8⁺ T cell gene-sets. Additionally, we observed that while Tx or anti-CD4 on their own drove enrichment of a CD8⁺ T cell exhaustion signature, the combination of Tx + anti-CD4 did not. Overall, this new analysis further supports our claim that only the Tx + anti-CD4 therapy drives a robust cytotoxic T cell program in the tumor.

Additionally, we have added flow cytometry data looking at percentages of terminally differentiated vs. stem-like CD8 T cells in the TdLN and tumor to further probe the T cell response. We did not observe any statistically significant differences among the various treatment groups

4. In Fig. 6A, low dose i.t. DT injection still dramatically reduced spleen Treg population by ~80%. This will induce systemic T cell activation that influences anti-tumor immunity indirectly.

We thank the reviewer for pointing this out, and we have included additional language to point out this caveat. Notably, this is also true for the anti-CD4 combination which will deplete spleen Tregs.

5. In the anti-CTLA-4 combination therapy experiment shown in Fig. 7, anti-CTLA-4 antibody treatment alone without Tx should be a critical control to be included.

We thank the reviewer for pointing out the lack of this control. We have added previously collected data from these studies where TA99 + anti-CTLA-4 was included as a negative control. We believe this is a sufficient control, and also note that the lack of anti-CTLA-4 efficacy in B16F10 tumors has been previously demonstrated in the literature (Reilley et al., JITC, 2019, PMID: 31771649 and a study from our own group, Stinson, Sheen et al., CCR, 2023, PMID: 37014656).

6. It has been shown that the anti-CTLA-4 therapy partially depleted intra-tumor Tregs (Simpson TR et al, JEM, 2013, PMID: 23897981.). This work is highly relevant to the current study and should be cited.

We thank the reviewer for pointing out the lack of discussion surrounding depletion of intratumoral Tregs by anti-CTLA-4 and we have added commentary on this (and relevant citations) to the discussion section.

REVIEWERS' COMMENTS

Reviewer #1 (Remarks to the Author):

I thank the authors for their very detailed response. My points were well addressed. In addition, the paper has further improved in quality by taking into account the points of the other two reviewers.

Reviewer #2 (Remarks to the Author):

- In response to my comments, the authors have added a satisfactory experiment comparing parental 4-1BB antibody and 4-1BB LAIR for agonist activity in presence and absence of collagen. This is useful. However, there are typo's in Extended data Figure 3 and the description in the Results section is not congruent with the content of the panels. This should be corrected.

- To convince me that 4-1BB-LAIR acts in the TME, the authors have added GSEA on their transcriptome data from day 6, which is certainly informative and helps to make this argument.

- On the topic of the need for CD4 T cells to support the CD8 T cell response, the authors have added some literature data on memory formation. However, there is also key literature on the need of CD4 T cell help for primary CTL effector function against cancer that is not yet cited. I think this is necessary to make sure that non-expert readers do not get the idea that total CD4 T cell depletion is a useful strategy in cancer immunotherapy (see Ahrends T et al. Immunity 2017, Nat Comm. 2019 and references therein).

-I feel that the title and other parts of the text should better reflect the finding on the synergy between boosting CTL priming by Treg depletion and boosting intratumoral CTL activity by local 4-1BB agonism. That would increase the perception of novelty.

Reviewer #3 (Remarks to the Author):

In the revised manuscript, the authors partially addressed the reviewer's concerns by performing additional analysis of the bulk RNA-seq data and adding more controls and related citations. On the other hand, the immune cell response to Tx/anti-CTLA-4 combination therapy still needs more detailed characterization. Concerns still remain on the value of bulk RNA-seq on all CD45+ immune cells isolated from TDLNs and tumors.

REVIEWERS' COMMENTS

Reviewer #1 (Remarks to the Author):

I thank the authors for their very detailed response. My points were well addressed. In addition, the paper has further improved in quality by taking into account the points of the other two reviewers.

We thank the reviewer for their kind comments regarding the updated manuscript.

Reviewer #2 (Remarks to the Author):

- In response to my comments, the authors have added a satisfactory experiment comparing parental 4-1BB antibody and 4-1BB LAIR for agonist activity in presence and absence of collagen. This is useful. However, there are typo's in Extended data Figure 3 and the description in the Results section is not congruent with the content of the panels. This should be corrected.

We thank the reviewer for finding our experiments satisfactory. However, we do not see the typo's the reviewer has pointed out in extended data figure 3 (now renamed supplementary figure 3). The figure demonstrates that 2.5F-Fc + anti-4-1BB-LAIR + anti-CD4 prolongs survival of MC38 tumor bearing mice more than either 2.5-F-Fc + anti-4-1BB-LAIR alone or anti-CD4 alone. This is consistent with the effect that we see with TA99 + anti-4-1BB-LAIR + anti-CD4 in B16F10 tumor bearing mice. As we note in the text, the complete response rate in 2.5F-Fc + anti-4-1BB-LAIR + anti-CD4 in MC38 tumor bearing mice is not as high as the complete response rate in TA99 + anti-4-1BB-LAIR + anti-CD4. We are happy to make any changes if specific typos that we have missed are pointed out.

If the reviewer is instead referring to Figure S1 (which contains the *in vitro* assay discussed in above this point), we did note a spelling error in the title of the graph, which we have corrected. Thank you for drawing our attention to this. In these assays the EC₅₀ values for both anti-4-1BB and anti-4-1BB-LAIR are within several nM of each other, notably with the LAIR fusion a slightly *less* potent agonist. We have modified the text to reflect that the agonistic potential with LAIR fusion is not unaffected, but rather is minimally affected.

- To convince me that 4-1BB-LAIR acts in the TME, the authors have added GSEA on their transcriptome data from day 6, which is certainly informative and helps to make this argument.

We thank the reviewer for their kind comments regarding the added GSEA.

- On the topic of the need for CD4 T cells to support the CD8 T cell response, the authors have added some literature data on memory formation. However, there is also key literature on the need of CD4 T cell help for primary CTL effector function against cancer that is not yet cited. I think this is necessary to make sure that non-expert readers do not get the idea that total CD4 T cell depletion is a useful strategy in cancer immunotherapy (see Ahrends T et al. Immunity 2017, Nat Comm. 2019 and references therein).

We have added recommended citations to the discussion section (specifically to the preexisting section that highlights why total CD4 T cell depletion is not a viable clinical approach).

-I feel that the title and other parts of the text should better reflect the finding on the synergy between boosting CTL priming by Treg depletion and boosting intratumoral CTL activity by local 4-1BB agonism. That would increase the perception of novelty.

We respectfully disagree and feel that the text and title sufficiently highlight the findings and accurately represent the data in the manuscript. We state directly at the end of the introduction that "This work suggests that locally retained 4-1BB agonist and antitumor antibody therapy can be highly efficacious when combined with modalities that enhance T cell priming, which can be restrained by TdLN Tregs." As we stated in the editor response, we feel that any reference to "depletion of lymph node Tregs" in the title would be misleading because, although we interpret our data to suggest that lymph node Treg depletion/inhibition is important for optimal responses, none of the therapies

employed deplete *only* tumor draining lymph node Tregs (as there are no currently available strategies for spatially restricted Treg depletion). Instead, our data in this manuscript provides strong rationale for the development of therapies that selectively deplete tumor draining lymph node Tregs, which we emphasize at the end of the discussion section.

Reviewer #3 (Remarks to the Author):

In the revised manuscript, the authors partially addressed the reviewer's concerns by performing additional analysis of the bulk RNA-seq data and adding more controls and related citations. On the other hand, the immune cell response to Tx/anti-CTLA-4 combination therapy still needs more detailed characterization. Concerns still remain on the value of bulk RNA-seq on all CD45+ immune cells isolated from TDLNs and tumors.

We respectfully maintain that additional single cell transcriptomic analysis consists of follow up work outside the scope of this manuscript.